# TINED: GNNs-to-MLPs by Teacher Injection and Dirichlet Energy Distillation

**Ziang Zhou** [1]  **Zhihao Ding** [1]  **Jieming Shi**[* 1]  **Qing Li** [1]  **Shiqi Shen** [2]

## Abstract

Graph Neural Networks (GNNs) are pivotal in graph-based learning, particularly excelling in node classification. However, their scalability is hindered by the need for multi-hop data during inference, limiting their application in latency-sensitive scenarios. Recent efforts to distill GNNs into multi-layer perceptrons (MLPs) for faster inference often underutilize the layer-level insights of GNNs. In this paper, we present TINED, a novel approach that distills GNNs to MLPs on a layer-by-layer basis using Teacher Injection and Dirichlet Energy Distillation techniques. We focus on two key operations in GNN layers: feature transformation (FT) and graph propagation (GP). We recognize that FT is computationally equivalent to a fully-connected (FC) layer in MLPs. Thus, we propose directly transferring teacher parameters from an FT in a GNN to an FC layer in the student MLP, enhanced by fine-tuning. In TINED, the FC layers in an MLP replicate the sequence of FTs and GPs in the GNN. We also establish a theoretical bound for GP approximation. Furthermore, we note that FT and GP operations in GNN layers often exhibit opposing smoothing effects: GP is aggressive, while FT is conservative. Using Dirichlet energy, we develop a DE ratio to measure these effects and propose Dirichlet Energy Distillation to convey these characteristics from GNN layers to MLP layers. Extensive experiments show that TINED outperforms GNNs and leading distillation methods across various settings and seven datasets. Source code are available at https://github.com/scottjiao/TINED_ICML25/.

---

[1]Department of Computing, The Hong Kong Polytechnic University, Hong Kong SAR, China [2]Wechat, Tencent, Beijing, China. Correspondence to: [*]Jieming Shi <jieming.shi@polyu.edu.hk>.

*Proceedings of the $42^{nd}$ International Conference on Machine Learning*, Vancouver, Canada. PMLR 267, 2025. Copyright 2025 by the author(s).

## 1. Introduction

Graph neural networks (GNNs) have delivered impressive outcomes in important applications (Hamilton et al., 2017; Kipf and Welling, 2017; Veličković et al., 2018). The power of GNNs is underpinned by the message passing framework that assimilates and refines node representations by considering their (multi-hop) neighborhood over graphs (Wu et al., 2020; Zhou et al., 2020). Nonetheless, the message passing is computationally demanding due to numerous nodes involved, posing significant challenges to deploy GNNs in latency-sensitive applications that require fast inference (Zhang et al., 2020; 2022a; Jia et al., 2020).

Recent studies attempt to combine the performance advantage of GNNs and the latency advantage of multi-layer perceptrons (MLPs) (Hu et al., 2021; Zhang et al., 2022b; Tian et al., 2022; Chen et al., 2021). Specifically, GLNN (Zhang et al., 2022b) distills teacher GNNs into student MLPs via soft labels. Then the student is deployed for fast inference to approximate the performance of GNNs without expensive message passing on graphs. NOSMOG (Tian et al., 2022) further considers graph structures, robustness, and node relations, while a new graph representation space is learned in (Yang et al., 2024). Current research often treats both teacher GNN and student MLP as monolithic model entities, focusing primarily on soft label outputs of the GNN for distillation, which overlooks the intrinsic knowledge contained within fine-grained GNN layers.

In this paper, we conduct an in-depth analysis to reveal important properties of the key operations in GNN layers and propose a novel method, TINED, distilling GNNs to MLPs layer-wise by Teacher INjection and Dirichlet Energy Distillation. The main ideas are explained below, and the detailed architecture of TINED is presented later.

**Main Idea of Teacher Injection.** A typical GNN layer includes two key operations: feature transformation (FT) and graph propagation (GP) (Kipf and Welling, 2017; Veličković et al., 2018; Hamilton et al., 2017; Liu et al., 2020; Zhu et al., 2021). GP aggregates neighbor representations of a node, while FT transforms node representations with learnable parameters. The valuable knowledge of a teacher GNN is preserved in the well-trained parameters of its FT and GP operations. Existing studies do not directly utilize these parameters (Zhang et al., 2022b). In Section 4.1, we recognize

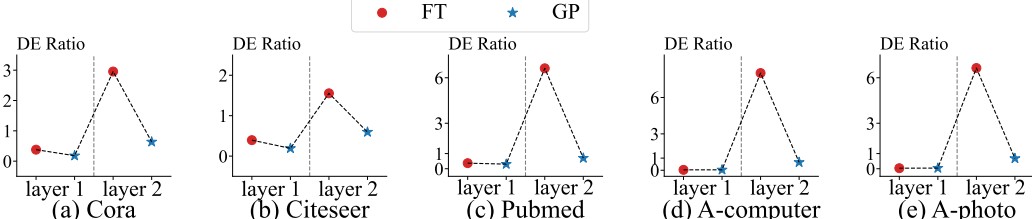

Figure 1: The DE ratios of FTs and GPs in the layers of GraphSAGE.

that *FT operations in GNNs share the same formulation as fully-connected (FC) layers in MLPs*, both transforming representations using learnable transformation matrices and activations. Thus, we propose to inject the parameters of an FT in a teacher GNN layer into an FC layer in student MLP, followed by fine-tuning for distillation. The GP operation in the GNN layer is emulated by another FC layer in the student. This allows TINED to directly transplant teacher knowledge into the student. Moreover, in TINED, the FC layers in the MLP mirror the order of their corresponding FTs and GPs in the GNN teacher, preserving layer-specific knowledge as much as possible. Theoretically, we prove an approximation bound between a GP in GNN and its corresponding FC layer in MLP, which depends on the eigenvalue of graph Laplacian matrix.

**Main Idea of Dirichlet Energy Distillation.** We further investigate the smoothing properties of FT and GP operations within GNN layers, since it is well-recognized that appropriate smoothing is crucial for GNNs (Chen et al., 2020a). Importantly, we observe that *the FT and GP operations in a GNN layer often exert opposing smoothing effects: GP aggressively smooths node embeddings, while FT is more restrained and can even diversify embeddings.* Using Dirichlet energy (Rusch et al., 2023), we propose a *DE ratio* measure to quantify whether an operation is conservative (large DE ratio) or aggressive (small DE ratio) in smoothing. Figure 1 shows the DE ratios of FTs and GPs in a trained 2-layer GNN teacher GraphSAGE on the experimental datasets. Within the same layer, the DE ratio of FT (red dot) is often larger than that of GP (blue star), indicating their opposing smoothing behaviors. Similar observations are made on other GNNs (see Appendix A.4). To distill these smoothing patterns from GNN to MLP layers, we design Dirichlet Energy Distillation in Section 4.2.

We conduct extensive experiments on benchmark datasets under various settings. Results show that TINED achieves superior performance and fast inference speed, compared with existing methods and various teacher GNNs. For example, on the Citeseer data, TINED improves a GNN teacher by 3.94%, MLPs by 15.93%, GLNN by up to 3.21%, and NOSMOG by 1.54%. TINED is 94× faster than its GNN teacher for inference. Our contributions are as follows:

- We propose TINED, a novel method to effectively distill fine-grained layer-wise knowledge from teacher GNNs into student MLPs.
- We develop a teacher injection technique to transplant the parameters of key operations from GNNs to MLPs. We provide a theoretical approximation analysis.
- We observe distinct smoothing effects of FT and GP operations in GNN layers and introduce Dirichlet energy distillation to impart these effects to student MLPs.
- Extensive experiments demonstrate that TINED achieves superior performance with various GNN teachers on widely-adopted benchmark datasets.

## 2. Related Work

GNNs leverage message-passing to aggregate neighborhood information for learning (Kipf and Welling, 2017; Veličković et al., 2018; Hamilton et al., 2017; Chen et al., 2020b; Klicpera et al., 2019). GCN (Kipf and Welling, 2017) introduces layer-wise propagation, GAT (Veličković et al., 2018) employs attention mechanisms, and APPNP (Klicpera et al., 2019) uses personalized PageRank. Chen et al. (2020b) mitigate over-smoothing with residual connections. Extensions include positional encoding (You et al., 2019; Wang et al., 2022; Li et al., 2020), few-shot learning (Zhou et al., 2023b; Dong et al., 2025a), and anomaly detection (Dong et al., 2025b; 2024; Ding et al., 2024). Specialized GNNs, such as heterogeneous GNNs (Zhou et al., 2023a; Zhang et al., 2019), address specific graph types. Despite their effectiveness, GNN inference is computationally expensive.

Knowledge Distillation has been applied in GNNs to accelerate inference while preserving effectiveness (Lee and Song, 2019; Yang et al., 2020; 2021; Tian et al., 2023; Feng et al., 2022). Previous studies have trained smaller student GNNs with fewer parameters than the large teacher GNNs, such as LSP (Yang et al., 2020), FreeKD (Feng et al., 2022) and TinyGNN (Yan et al., 2020) which still rely on time-intensive message passing. GFKD (Deng and Zhang, 2021) performs distillation at graph-level instances, while we target nodes within a single graph. Recent studies (Tian et al., 2022; Yang et al., 2024; Zhang et al., 2022b; Hu et al., 2021; Zheng et al., 2022; Wu et al., 2023a;b) develop MLP-based

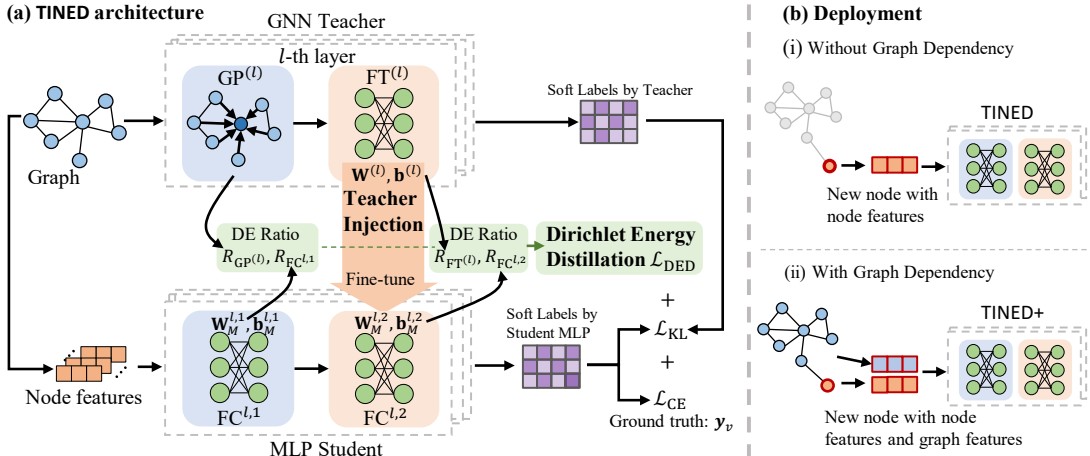

Figure 2: (a) TINED with Teacher Injection and Dirichlet Energy Distillation; (b) Inference settings

student models without message passing. GLNN (Zhang et al., 2022b) trains an MLP student using node features as input and soft labels from a GNN teacher as targets. NOSMOG (Tian et al., 2022) incorporates graph structure, adversarial feature augmentation, and node similarity relations into distillation. VQGraph learns a structure-aware tokenizer to encode node substructures (Yang et al., 2024), during which teacher is also trained. As mentioned, the intrinsic knowledge in well-trained GNN layers is insufficiently utilized by existing studies.

There exist works in other orthogonal domains, also observing the benefits of distilling layer-wise teacher structures, such as language models (Liang et al., 2023), natural language processing (Liang et al., 2023; Chang et al., 2022; Jiao et al., 2020) and computer vision (Li et al., 2019; Hinton et al., 2015). In this work, we fill the gap to develop layer-wise distillation method for GNNs. In addition, Han et al. (2023) work on a different problem that uses pre-trained MLPs for GNN training acceleration.

## 3. Preliminaries

**Notations.** A graph $\mathcal{G} = (\mathcal{V}, \mathcal{E}, \mathbf{X})$ consists of a node set $\mathcal{V}$ with $n$ nodes, i.e., $|\mathcal{V}| = n$, an edge set $\mathcal{E}$ of size $|\mathcal{E}| = m$, and $\mathbf{X} \in \mathbb{R}^{n \times d_c}$ for $d_c$-dimensional features of all nodes. Let $\mathbf{A}$ be the adjacency matrix of $\mathcal{G}$, where $\mathbf{A}_{u,v} = 1$ if edge $(u, v) \in \mathcal{E}$, and 0 otherwise. A node $v$ has a neighbor set $\mathcal{N}(v) = \{u | (u, v) \in \mathcal{E}\}$, and the degree of node $v$ is $|\mathcal{N}(v)|$. Degree matrix $\mathbf{D}$ is a diagonal matrix with $\mathbf{D}_{v,v} = |\mathcal{N}(v)|$. In node classification task, the prediction targets are $\mathbf{Y} \in \mathbb{R}^{n \times K}$, where $K$ is the number of classes, and row $\mathbf{y}_v$ is a $K$-dim one-hot vector indicating the ground-truth class of node $v$. A model predicts the class probabilities of node $v$. In $\mathcal{G}$, we use superscript $^L$ to mark the labeled nodes (i.e., $\mathcal{V}^L$, $\mathbf{X}^L$, and $\mathbf{Y}^L$), and superscript $^U$ to mark the unlabeled nodes (i.e., $\mathcal{V}^U$, $\mathbf{X}^U$, and $\mathbf{Y}^U$).

**GNNs.** Let $\mathbf{H}^{(l)} \in \mathbb{R}^{n \times d_l}$ be the output node embedding matrix of the $l$-th GNN layer, with each row $\mathbf{h}_v^{(l)}$ being the representation of node $v$ in $\mathcal{V}$. Most GNNs fit under the message-passing framework with feature transformation (FT) and graph propagation (GP) operations (Zhu et al., 2021). The $l$-th layer usually comprises operations $\text{FT}^{(l)}$ and $\text{GP}^{(l)}$ to get $\mathbf{H}^{(l)}$ in Eq. (1). $\text{GP}^{(l)}$ aggregates the $(l-1)$-th layer representations $\mathbf{H}^{(l-1)}$ over $\mathcal{G}$, and $\text{FT}^{(l)}$ transforms representations to get $\mathbf{H}^{(l)}$. Different GNNs may vary in how they composite $\text{FT}^{(l)}$ and $\text{GP}^{(l)}$.

$$\mathbf{H}^{(l)} = \text{FT}^{(l)}\left(\text{GP}^{(l)}\left(\mathbf{H}^{(l-1)}, \mathcal{G}\right)\right) \quad (1)$$

**MLPs.** An MLP is composed of multiple fully-connected (FC) layers. The $l$-th FC layer converts an embedding $\mathbf{h}^{(l-1)}$ to $\mathbf{h}^{(l)}$ via a transformation matrix $\mathbf{W}_M^{(l)}$ in Eq. (2).

$$\mathbf{h}^{(l)} = \sigma(\mathbf{h}^{(l-1)}\mathbf{W}_M^{(l)} + \mathbf{b}^{(l)}), \quad (2)$$

where $\sigma$ is an activation and $\mathbf{b}^{(l)}$ is bias.

**GNNs-to-MLPs Distillation.** Given a pre-trained GNN teacher, the goal is to train a cost-effective student MLP to predict $\hat{\mathbf{y}}_v$ for node $v$, utilizing ground-truth labels $\mathbf{y}_v$ of labeled nodes $v \in \mathcal{V}^L$ and soft labels $\mathbf{z}_v$ produced by teacher GNN for all $v \in \mathcal{V}$. The training objective is formulated below (Zhang et al., 2022b). Note that the student is trained by both ground truth and soft labels from the teacher.

$$\mathcal{L} = \sum_{v \in \mathcal{V}^L} \mathcal{L}_{CE}(\hat{\mathbf{y}}_v, \mathbf{y}_v) + \lambda \sum_{v \in \mathcal{V}} \mathcal{L}_{KL}(\hat{\mathbf{y}}_v, \mathbf{z}_v), \quad (3)$$

where $\mathcal{L}_{CE}$ is the cross-entropy loss by comparing the student MLP predictions $\hat{\mathbf{y}}_v$ with the ground truth $\mathbf{y}_v$, and $\mathcal{L}_{KL}$ is the KL-divergence loss between the student MLP predictions $\hat{\mathbf{y}}_v$ and the soft labels $\mathbf{z}_v$ from teacher, and weight $\lambda$ balances the two losses.

## 4. The TINED Method

In Eq. (3), existing studies primarily use GNN soft labels and ground truth labels for distillation. Our approach TINED

expands this to include model parameters and layer-specific properties of teacher GNNs for distilling knowledge into MLPs, as illustrated in Figure 2. As shown in Figure 2(a), for the Teacher Injection technique, we identify that the $\mathrm{FT}^{(l)}$ operation of the $l$-th GNN layer essentially conducts the same computation as an FC layer in an MLP, and thus, we inject the parameters of $\mathrm{FT}^{(l)}$ into an FC layer $\mathrm{FC}^{l,2}$ of the student MLP. The injected parameters are fine-tuned during the distillation to ensure controlled adaption. To emulate the $\mathrm{GP}^{(l)}$ operation in the $l$-th GNN layer, we employ another FC layer $\mathrm{FC}^{l,1}$ in the student MLP. We establish a theoretical approximation bound between $\mathrm{GP}^{(l)}$ and $\mathrm{FC}^{l,1}$. Then as in Figure 2(a), we use Dirichlet energy to develop DE ratios $R_{\mathrm{GP}^{(l)}}$ and $R_{\mathrm{FT}^{(l)}}$ to quantify the smoothing effects of $\mathrm{GP}^{(l)}$ and $\mathrm{FT}^{(l)}$ in the $l$-th GNN layer, and design Dirichlet Energy Distillation with a $\mathcal{L}_{DED}$ loss to ensure the DE ratios $R_{\mathrm{FC}^{l,2}}$ and $R_{\mathrm{FC}^{l,1}}$ of the student MLP's FC layers $\mathrm{FC}^{l,1}$ and $\mathrm{FC}^{l,2}$ preserve the smoothing effects of $\mathrm{GP}^{(l)}$ and $\mathrm{FT}^{(l)}$ from the teacher GNN.

There are two typical inference deployment settings as depicted in Figure 2(b). TINED can operate without graph dependency for inference (Zhang et al., 2022b), which is useful when the graph structure is unavailable or when new unseen nodes lack graph connections; TINED can also work with graph structures when allowed (Tian et al., 2022). Following (Zhang et al., 2022b; Tian et al., 2022), we use GraphSAGE (Hamilton et al., 2017) with GCN aggregation as the teacher GNN to explain TINED. Our experiments include results over different teacher GNNs. In Appendix A.3, we explain how to use TINED for various GNNs.

### 4.1. Teacher Injection

The idea is to inject the parameters of certain operations of GNNs into MLPs, thereby directly transferring the teacher's knowledge to the student. The GNN operations to be injected should be compatible with MLP in Eq. (2), i.e., with the same formulation. As shown in Eq. (1), a GNN layer consists of GP and FT operations. In what follows, we show that the FT operations of GNNs can be injected into student MLPs. We use GraphSAGE as an example, while the analysis on other GNNs as teacher is in Appendix A.3. The $l$-th layer of GraphSAGE with $\mathrm{GP}^{(l)}$ and $\mathrm{FT}^{(l)}$ operations is

$$\mathrm{GP}^{(l)}: \tilde{\mathbf{h}}_v^{(l)} = \mathrm{CAT}\left(\mathbf{h}_v^{(l-1)}, \mathrm{AGG}^{(l)}\left(\{\mathbf{h}_u^{(l-1)}, \forall u \in \mathcal{N}(v)\}\right)\right),$$
$$\mathrm{FT}^{(l)}: \mathbf{h}_v^{(l)} = \sigma\left(\tilde{\mathbf{h}}_v^{(l)} \cdot \mathbf{W}^{(l)} + \mathbf{b}^{(l)}\right).$$
(4)

In Eq. (4), $\mathrm{GP}^{(l)}$ operation has an aggregator $\mathrm{AGG}^{(l)}$ to combine the $(l-1)$-th representations $\mathbf{h}_u^{(l-1)}$ of $v$'s neighbors and then concatenates (CAT) them with $\mathbf{h}_v^{(l-1)}$ to get $\tilde{\mathbf{h}}_v^{(l)}$. In $\mathrm{FT}^{(l)}$ operation, a learnable transformation matrix $\mathbf{W}^{(l)}$ is applied to $\tilde{\mathbf{h}}_v^{(l)}$, followed by an activation function to yield the $l$-th layer representation $\mathbf{h}_v^{(l)}$ of $v$.

Observe that (i) $\mathrm{FT}^{(l)}$ operates independently of the graph, whereas $\mathrm{GP}^{(l)}$ requires the graph; (ii) $\mathrm{FT}^{(l)}$ in Eq. (4) has the same formulation as an FC layer of an MLP in Eq. (2). Therefore, to approximate the $l$-th GNN layer, we employ two FC layers $\mathrm{FC}^{l,1}$ and $\mathrm{FC}^{l,2}$ in the student MLP. In Eq. (5), the $\mathrm{FC}^{l,1}$ layer of the student MLP is to approximate $\mathrm{GP}^{(l)}$ in Eq. (4), while the $\mathrm{FC}^{l,2}$ layer approximates $\mathrm{FT}^{(l)}$ in Eq. (4), with teacher injection to set $\mathbf{W}_M^{l,2} = \mathbf{W}^{(l)}$ and $\mathbf{b}_M^{l,2} = \mathbf{b}^{(l)}$, transferring the parameters of $\mathrm{FT}^{(l)}$ to $\mathrm{FC}^{l,2}$. $\mathbf{s}_v^{(l)}$ is the embedding of node $v$ generated by the student.

$$\mathrm{FC}^{l,1} \text{ for } \mathrm{GP}^{(l)}: \hat{\mathbf{s}}_v^{(l)} = \sigma\left(\mathbf{s}_v' \mathbf{W}_M^{l,1} + \mathbf{b}_M^{l,1}\right),$$
$$\mathrm{FC}^{l,2} \text{ for } \mathrm{FT}^{(l)}: \mathbf{s}_v^{(l)} = \sigma\left(\hat{\mathbf{s}}_v \mathbf{W}_M^{l,2} + \mathbf{b}_M^{l,2}\right),$$
(5)

where $\mathbf{W}_M^{l,2} = \mathbf{W}^{(l)}$ and $\mathbf{b}_M^{l,2} = \mathbf{b}^{(l)}$, and $\mathbf{W}^{(l)}$ and $\mathbf{b}^{(l)}$ are from Eq. (4).

Different from existing methods, TINED distills knowledge from GNNs to MLPs on a per-layer basis. Directly injecting GNN parameters into the MLP is anticipated to improve the effectiveness. For layer-wise GNNs with $T$ layers, the resulting MLP will have $2T$ FC layers. Typically, $T$ is small, e.g., 2 or 3, and the efficiency impact on inference between MLPs with 2 and 4 layers is minimal (1.45ms vs. 1.63ms, as reported in (Zhang et al., 2022b), compared to 153ms by GraphSAGE). Furthermore, our experiments show that TINED can achieve a favorable tradeoff between effectiveness and efficiency for inference. For decoupled GNNs, e.g., APPNP (Klicpera et al., 2019), with FTs and GPs decoupled, the teacher injection is also applicable with a small number of FC layers (see Appendix A.3).

In Theorem 4.1, we provide a theoretical approximation bound between $\mathrm{GP}^{(l)}$ in the $l$-th layer of teacher GNN and its corresponding $\mathrm{FC}^{l,1}$ in student MLP, serving as an attempt to establish the relationship between GNNs and MLPs for knowledge distillation. The proof is in Appendix A.2.

**Theorem 4.1.** *For a sparse matrix $\mathbf{L} \in \mathbb{R}^{n \times n}$ and a feature matrix $\mathbf{H} \in \mathbb{R}^{n \times d}$ with $rank(\mathbf{H}) = d$, there exists a transformation matrix $\mathbf{W}^*$ to approximate $\mathbf{LH}$ by $\mathbf{HW}^*$ with relative error*

$$\frac{||\mathbf{LH} - \mathbf{HW}^*||_F}{||\mathbf{H}||_F} \leq \lambda_{\max}(\mathbf{L}),$$

*where $|| \cdot ||_F$ is the Frobenius norm and $\lambda_{\max}(\mathbf{L})$ is the largest eigenvalue of laplacian $\mathbf{L}$.*

Specifically, Laplacian matrix $\mathbf{L}$ represents graph topology used in $\mathrm{GP}^{(l)}$, $\mathbf{LH}$ represents the output of $\mathrm{GP}^{(l)}$, while $\mathbf{HW}^*$ represents the output of $\mathrm{FC}^{l,1}$ and $\mathbf{HW}^*$ approximates $\mathbf{LH}$. The relative error is upper-bounded by the largest eigenvalue $\lambda_{\max}(\mathbf{L})$ of $\mathbf{L}$, regardless of graph size.

The parameters of FC layers $\mathrm{FC}^{l,2}$ in Eq. (5) are sourced from the $\mathrm{FT}^{(l)}$ operations of teacher GNN. To ensure these

FC layers contribute to the student MLP's training in a regulated manner, we introduce a gradient modifier $\eta$ to fine-tune the injected parameters when updating the gradients,

$$\begin{aligned} \hat{\nabla}\mathbf{W}_M^{l,2} &= \eta\nabla\mathbf{W}_M^{l,2} \\ \hat{\nabla}\mathbf{b}_M^{l,2} &= \eta\nabla\mathbf{b}_M^{l,2}, \end{aligned} \tag{6}$$

where $\eta$ is a hyperparameter to control fine-tuning, while $\nabla$ represents gradient.

In other words, during training, the update of the parameters in $\text{FC}^{l,2}$ based on gradient optimization after backward propagation is adjusted by $\eta$ parameter.

### 4.2. Dirichlet Energy Distillation

It is widely recognized that an appropriate degree of smoothing is crucial for the efficacy of GNNs. Here we make an interesting observation that, within the $l$-th GNN layers, $\text{GP}^{(l)}$ operations tend to aggressively smooth node embeddings, while $\text{FT}^{(l)}$ operations often apply conservative smoothing or even diversify embeddings. These distinct layer-level smoothing behaviors in teacher GNNs should be captured in the student MLPs.

To achieve this, we introduce Dirichlet Energy Distillation. Dirichlet energy (Rusch et al., 2023) is a measure commonly used to quantify the degree of smoothing in embeddings by evaluating node pair distances. Definition 4.2 defines the Dirichlet energy $E(\mathbf{H})$ of a node embedding matrix $\mathbf{H}$.

**Definition 4.2.** Given a node embedding matrix $\mathbf{H} \in \mathbb{R}^{n \times d}$, learned from either the GNN teacher or the MLP student at a certain layer, the Dirichlet energy of $\mathbf{H}$ is

$$E(\mathbf{H}) = \frac{1}{n} \cdot \text{tr}(\mathbf{H}^\top \mathbf{L}\mathbf{H}), \tag{7}$$

where $\text{tr}(\cdot)$ is the trace of a matrix, and $\mathbf{L} = \mathbf{D} - \mathbf{A}$ is the Laplacian matrix of graph $\mathcal{G}$.

A lower Dirichlet energy value $E(\mathbf{H})$ suggests that the embeddings in $\mathbf{H}$ are smooth, whereas a higher value indicates diversity among the embeddings. For an operation $op$ that processes $\mathbf{H}$ to output $op(\mathbf{H})$, if $E(op(\mathbf{H})) < E(\mathbf{H})$, then the operation $op$ is smoothing the embeddings; on the other hand, if $E(op(\mathbf{H})) > E(\mathbf{H})$, $op$ diversifies them.

We define Dirichlet energy ratio $R_{op}$ (DE ratio) as follows.

**Definition 4.3** (DE ratio). The DE ratio $R_{op}$ of an operation $op$ is the Dirichlet energy of its output $E(op(\mathbf{H}))$ over the Dirichlet energy of its input $E(\mathbf{H})$, $R_{op} = \frac{E(op(\mathbf{H}))}{E(\mathbf{H})}$.

On a trained teacher GraphSAGE with 2 layers, in each layer, we calculate the DE ratios $R_{\text{FT}^{(l)}}$ and $R_{\text{GP}^{(l)}}$ of $\text{FT}^{(l)}$ and $\text{GP}^{(l)}$ operations respectively. We repeat 10 times and report the average DE ratios in Figure 1, over the benchmark datasets used in experiments.

While the overall Dirichlet energy of embeddings is decreasing in GraphSAGE, in a specific $l$-th layer shown in Figure 1, we make the following two consistent observations about DE ratio. (i) Within the same layer for $l = 1, 2$, the DE ratio $R_{\text{FT}^{(l)}}$ of $\text{FT}^{(l)}$ consistently exceeds the DE ratio $R_{\text{GP}^{(l)}}$ of $\text{GP}^{(l)}$, demonstrating that $\text{GP}^{(l)}$ operation actively smooths embeddings, whereas $\text{FT}^{(l)}$ operation is relatively conservative for smoothing. (ii) At $l = 2$, DE ratio $R_{\text{FT}^{(l)}}$ even surpasses 1, indicating that in this layer, $\text{FT}^{(l)}$ acts to diversify embeddings rather than smoothing them. These trends are consistently observed across layer-wise GNNs, including GCN and GAT in Appendix A.4.

Figure 1 reveals that FTs and GPs can have opposing effects on smoothing. Recall that we associate $\text{FT}^{(l)}$ and $\text{GP}^{(l)}$ of the $l$-th GNN layer with $\text{FC}^{l,2}$ and $\text{FC}^{l,1}$ layers in the student MLP in Eq. (5). The proposed Dirichlet Energy Distillation technique aims to encapsulate the distinct smoothing behaviors of $\text{FT}^{(l)}$ and $\text{GP}^{(l)}$ into $\text{FC}^{l,2}$ and $\text{FC}^{l,1}$ respectively, thereby transferring teacher GNN's knowledge of smoothing effects to student MLP.

Specifically, the loss of the $l$-th GNN layer for Dirichlet Energy Distillation, $\mathcal{L}_{DED}^{(l)}$, is the sum of the squares of the difference between DE ratios $R_{\text{GP}^{(l)}}$ of $\text{GP}^{(l)}$ and $R_{\text{FC}^{l,1}}$ of the $\text{FC}^{l,1}$ layer, and the difference between $R_{\text{FT}^{(l)}}$ of $\text{FT}^{(l)}$ and $R_{\text{FC}^{l,2}}$ of the $\text{FC}^{l,2}$ layer. The total DED loss $\mathcal{L}_{DED}$ is the sum of $\mathcal{L}_{DED}^{(l)}$ for all GNN layers $l = 1, .., T$.

$$\begin{aligned} \mathcal{L}_{DED}^{(l)} &= \left(R_{\text{GP}^{(l)}} - R_{\text{FC}^{l,1}}\right)^2 + \left(R_{\text{FT}^{(l)}} - R_{\text{FC}^{l,2}}\right)^2 \\ \mathcal{L}_{DED} &= \sum_{l=1}^{T} \mathcal{L}_{DED}^{(l)}. \end{aligned} \tag{8}$$

The final objective function $\mathcal{L}$ of our method TINED is the weighted combination of ground truth cross-entropy loss $\mathcal{L}_{CE}$, soft label distillation loss $\mathcal{L}_{KL}$, and Dirichlet Energy Distillation loss $\mathcal{L}_{DED}$:

$$\mathcal{L} = \sum_{v \in \mathcal{V}^L} \mathcal{L}_{CE}(\hat{\mathbf{y}}_v, \mathbf{y}_v) + \lambda \sum_{v \in \mathcal{V}} \mathcal{L}_{KL}(\hat{\mathbf{y}}_v, \mathbf{z}_v) + \beta \sum_{l=1}^{T} \mathcal{L}_{DED}^{(l)}, \tag{9}$$

where $\lambda$ and $\beta$ are weights for balancing the loss functions.

## 5. Experiments

### 5.1. Experiment Settings

**Datasets.** We use 7 widely used public benchmark datasets, including Cora, Citeseer, Pubmed, A-computer, A-photo (Zhang et al., 2022b; Yang et al., 2021), and Arxiv and Products (Hu et al., 2020) that are two large OGB datasets, to evaluate our method and baselines. Table 10 in Appendix A.1 provides the data statistics and splits.

**Teacher Architectures.** Following (Zhang et al., 2022b;

Table 1: Under the transductive(*tran*) setting, the accuracy results of the teacher and all methods without or with graph dependency for inference on $\mathcal{V}^U$ (higher is better). The best in each category is in bold. OOM means out-of-memory.

| Datasets | Teacher | Without Graph Dependency | | | | | | With Graph Dependency | | |
|---|---|---|---|---|---|---|---|---|---|---|
| | SAGE | MLP | FFG2M | KRD | GLNN | GLNN* | TINED | NOSMOG | NOSMOG* | TINED+ |
| Cora | 80.64±1.57 | 59.18±1.60 | 82.38±1.41 | 82.27±1.31 | 80.26±1.66 | 81.31±1.62 | **82.63±1.57** | 83.04±1.26 | 82.27±1.75 | **83.70±1.02** |
| Citeseer | 70.49±1.53 | 58.50±1.86 | 72.85±1.59 | 72.84±1.70 | 71.22±1.50 | 72.38±1.40 | **74.43±1.53** | 73.78±1.54 | 73.85±2.27 | **75.39±1.59** |
| Pubmed | 75.56±2.06 | 68.39±3.09 | 76.56±3.41 | 77.01±3.11 | 75.59±2.46 | 76.95±2.72 | **77.09±2.14** | 77.34±2.36 | 76.79±2.65 | **77.75±3.14** |
| A-computer | 82.82±1.37 | 67.62±2.21 | 83.67±1.04 | 82.87±0.87 | 82.71±1.18 | 83.64±1.13 | **85.18±1.12** | 84.04±1.01 | 84.33±1.14 | **84.82±1.58** |
| A-photo | 90.85±0.87 | 77.29±1.79 | 93.18±0.87 | 92.82±0.74 | 91.95±1.04 | 92.99±0.63 | **93.97±0.53** | 93.36±0.69 | 93.57±0.48 | **94.05±0.39** |
| Arxiv | 70.73±0.35 | 55.67±0.24 | 58.51±0.35 | 59.26±0.51 | 63.75±0.48 | 63.78±0.69 | **64.44±0.72** | **71.65±0.29** | 71.17±0.60 | 71.52±0.34 |
| Products | 77.17±0.32 | 60.02±0.10 | OOM | OOM | 63.71±0.31 | 65.56±0.26 | **69.48±0.25** | 78.45±0.38 | 78.47±0.28 | **78.59±0.28** |

Tian et al., 2022), for the main results, GraphSAGE (Hamilton et al., 2017) with GCN aggregation is used as the teacher model. We also conduct experiments of different GNN teachers, e.g., GCN, GAT and APPNP in Section 5.5. The details of teachers and hyperparameter settings are in A.7.

**Methods.** TINED does not require graph dependency for inference, similar to GLNN (Zhang et al., 2022b). When graph dependency is permissible, we enhance it to create TINED+, akin to NOSMOG (Tian et al., 2022), to incorporate graph structural information. Baselines GLNN and NOSMOG use their original layer and hidden dimension configurations as per their papers. Additionally, we configure them with the same layer and hidden dimension settings as ours, dubbed as GLNN* and NOSMOG*, for evaluation. We further compare with KRD and FFG2M (Wu et al., 2023b;a), as well as with MLPs and the GraphSAGE teacher.

**Transductive and Inductive Settings.** Two settings are considered (Zhang et al., 2022b; Tian et al., 2022): transductive (*tran*) setting, and production (*prod*) setting with both inductive and transductive evaluations (*ind & tran*).

*tran*: A model is trained on $\mathcal{G}$, $\boldsymbol{X}$, and $\boldsymbol{Y}^L$, and soft labels $\mathbf{z}_v$ of all nodes in $\mathcal{V}$ are used for knowledge distillation. Inference evaluation is conducted on the nodes in $\mathcal{V}^U$.

*prod* (*ind & tran*): we randomly select out 20% of nodes from the unlabeled nodes $\mathcal{V}^U$, dividing $\mathcal{V}^U$ into disjoint inductive (unobserved) subset and observed subset, $\mathcal{V}^U = \mathcal{V}^U_{obs} \sqcup \mathcal{V}^U_{ind}$. Node features and labels are partitioned into disjoint sets, i.e. $\boldsymbol{X} = \boldsymbol{X}^L \sqcup \boldsymbol{X}^U_{obs} \sqcup \boldsymbol{X}^U_{ind}$, and $\boldsymbol{Y} = \boldsymbol{Y}^L \sqcup \boldsymbol{Y}^U_{obs} \sqcup \boldsymbol{Y}^U_{ind}$. Let $\mathcal{G}_{obs}$ be the graph induced from $\mathcal{G}$, with edges connecting nodes in $\mathcal{V}^L \sqcup \mathcal{V}^U_{obs}$. A model is trained on $\mathcal{G}_{obs}$, $\boldsymbol{X}^L$, $\boldsymbol{X}^U_{obs}$, and $\boldsymbol{Y}^L$. Soft labels in subsets $\mathcal{V}^L \sqcup \mathcal{V}^U_{obs}$ are used for distillation. In this *prod* (*ind & tran*) setting, inference is evaluated on $\mathcal{V}^U_{ind}$ for *ind* and $\mathcal{V}^U_{obs}$ for *tran* (which is different from the *tran* setting over $\mathcal{V}^U$ explained above). The overall *prod* performance is the weighted sum of *ind* performance on $\mathcal{V}^U_{ind}$ and *tran* on $\mathcal{V}^U_{obs}$, with weights proportaional to the size of $\mathcal{V}^U_{ind}$ and $\mathcal{V}^U_{obs}$.

**Evaluation.** Following (Tian et al., 2022; Zhang et al., 2022b), we present the mean and standard deviation of performance results from 10 trials, each with a unique random seed. A method is evaluated by accuracy with the best model chosen on validation data and tested on test data.

### 5.2. Performance in Transductive Setting

Table 1 reports the results under the transductive setting with inference on node set $\mathcal{V}^U$. Table 1 can be directly comparable to those reported in literature (Tian et al., 2022; Zhang et al., 2022b; Hu et al., 2020; Yang et al., 2021). In Table 1, GNNs-to-MLPs methods are in two categories: with or without graph dependency for inference. Baselines GLNN and NOSMOG use their original student MLP configurations for layers and hidden dimensions as recommended in their respective papers. GLNN* and NOSMOG* (with an asterisk *) adopt the same configurations as our model for a fair comparison. TINED surpasses all baselines without graph dependency, often with a significant margin. For example, on the large OGB Products dataset, TINED achieves gains of 9.42%, 5.77%, and 3.92% over MLP, GLNN, and GLNN*, while baselines KRD and FFG2M are OOM. On Citeseer, TINED achieves 74.43% accuracy, improving by 15.93%, 3.21%, 2.05%, 1.59%, and 1.58% over MLP, GLNN, GLNN*, KRD, and FFG2M. The performance of TINED validates the power of the proposed techniques to preserve layer-level knowledge into MLPs. Similar to GLNN, TINED surpasses the teacher on the first five data, while being effective on the last two. When graph dependency is allowed, TINED+ excels NOSMOG and NOSMOG* on most datasets, except Arxiv where the performance is comparable. On Citeseer, TINED+ achieves 75.39% accuracy, 1.54% higher than NOSMOG*. Moreover, all methods with graph dependency, including TINED+ and NOSMOG, perform better than those without graph, and TINED+ excels the teacher model on all datasets, indicating the extensibility of TINED+ to consider graph structures for inference.

### 5.3. Performance in Production Setting with *ind & tran*

We then conduct experiments under the *prod* (*ind & tran*) setting with results in Table 2. Note that the *tran* results in Table 2 are over $\mathcal{V}^U_{obs}$, different from the results in Table

Table 2: In *prod* (*ind* & *tran*) setting, the accuracy results of the teacher and all methods without or with graph dependency for online inference are presented. *ind* indicates the results on $\mathcal{V}_{ind}^U$, *tran* indicates the results on $\mathcal{V}_{obs}^U$, and *prod* indicates the weighted sum of the performance of both *ind* and *tran* with weights proportional to the size of $\mathcal{V}_{ind}^U$ and $\mathcal{V}_{obs}^U$. The best result in each category is in bold. The teacher's performance surpassing the best result in at least one category is italicized.

| Datasets | Eval | Teacher SAGE | Without Graph Dependency MLP | FFG2M | KRD | GLNN | GLNN* | TINED | With Graph Dependency NOSMOG | NOSMOG* | TINED+ |
|---|---|---|---|---|---|---|---|---|---|---|---|
| Cora | prod | 79.53 | 59.18 | 78.60 | 75.74 | 77.82 | 78.14 | **78.90** | **81.02** | 80.30 | 80.77 |
| | ind | *81.03±1.71* | 59.44±3.36 | 72.02±1.43 | 70.26±1.94 | 73.21±1.50 | 73.58±1.42 | **74.38±1.28** | 81.36±1.53 | 80.98±2.39 | **81.50±2.54** |
| | tran | 79.16±1.60 | 59.12±1.49 | 80.01±1.41 | 77.11±1.44 | 78.97±1.56 | 79.65±1.45 | **80.04±1.50** | **80.93±1.65** | 80.13±1.64 | 80.59±1.45 |
| Citeseer | prod | 68.06 | 58.49 | 71.89 | 71.38 | 69.08 | 70.91 | **72.29** | 70.60 | 71.33 | **73.58** |
| | ind | 69.14±2.99 | 59.31±4.56 | 69.75±3.16 | 69.78±3.04 | 68.48±2.38 | 71.10±1.50 | **72.68±1.97** | 70.30±2.30 | 72.35±2.99 | **74.20±1.67** |
| | tran | 67.79±2.80 | 58.29±1.94 | 72.12±2.69 | 71.77±2.81 | 69.23±2.39 | 70.86±1.66 | **72.20±1.66** | 70.67±2.25 | 71.07±1.85 | **73.43±1.63** |
| Pubmed | prod | 74.77 | 68.39 | 73.98 | **76.00** | 74.67 | 75.21 | 75.79 | 75.82 | 75.57 | **75.90** |
| | ind | 75.07±2.89 | 68.28±3.25 | 73.49±7.91 | 75.17±3.11 | 74.52±2.95 | 74.83±2.83 | **75.64±3.02** | 75.87±3.32 | 75.49±2.96 | **76.30±2.95** |
| | tran | 74.70±2.33 | 68.42±3.06 | 74.10±7.78 | **76.20±3.00** | 74.70±2.75 | 75.30±2.70 | 75.83±2.81 | 75.80±3.06 | 75.58±2.83 | **75.80±2.88** |
| A-computer | prod | 82.73 | 67.62 | 82.69 | 81.17 | 82.10 | 83.23 | **84.46** | 83.85 | 85.02 | **85.08** |
| | ind | *82.83±1.51* | 67.69±2.20 | 80.52±1.56 | 79.15±1.82 | 80.27±2.11 | 81.10±1.49 | **82.83±1.45** | 84.36±1.57 | 85.23±1.51 | **85.45±1.60** |
| | tran | 82.70±1.34 | 67.60±2.23 | 83.23±1.36 | 81.67±1.92 | 82.56±1.80 | 83.77±1.36 | **84.87±1.38** | 83.72±1.44 | 84.97±1.44 | **84.98±1.32** |
| A-photo | prod | 90.45 | 77.29 | 92.35 | 91.84 | 91.34 | 91.54 | **93.38** | 92.47 | 92.00 | **93.12** |
| | ind | 90.56±1.47 | 77.44±1.50 | 90.70±0.76 | 90.04±1.12 | 89.50±1.12 | 89.35±0.89 | **91.96±0.72** | 92.61±1.09 | 92.93±0.96 | **93.27±0.85** |
| | tran | 90.42±0.68 | 77.25±1.90 | 92.77±0.24 | 92.29±0.63 | 91.80±0.49 | 92.09±0.71 | **93.74±0.51** | 92.44±0.51 | 91.77±0.69 | **93.08±0.68** |
| Arxiv | prod | *70.69* | 55.35 | 59.60 | 59.32 | 63.50 | **64.17** | 63.24 | 70.90 | 70.95 | **71.22** |
| | ind | *70.69±0.58* | 55.29±0.63 | 57.02±0.43 | 57.32±0.31 | 59.04±0.46 | 58.73±0.46 | **59.79±0.46** | 70.09±0.55 | 70.12±0.39 | **70.42±0.35** |
| | tran | *70.69±0.39* | 55.36±0.34 | 60.24±0.23 | 59.82±0.27 | 64.61±0.15 | **65.53±0.30** | 64.10±0.38 | 71.10±0.34 | 71.16±0.15 | **71.43±0.19** |
| Products | prod | *76.93* | 60.02 | OOM | OOM | 63.47 | 68.48 | **69.35** | 77.33 | 78.25 | **78.91** |
| | ind | *77.23±0.24* | 60.02±0.09 | OOM | OOM | 63.38±0.33 | 68.13±0.20 | **68.68±0.27** | 77.02±0.19 | 78.52±0.22 | **79.31±0.29** |
| | tran | *76.86±0.27* | 60.02±0.11 | OOM | OOM | 63.49±0.31 | 68.57±0.20 | **69.52±0.27** | 77.41±0.21 | 78.18±0.23 | **78.81±0.29** |

1 on $\mathcal{V}^U$ in Section 5.2. In Table 2, TINED and TINED+ can outperform the teacher model and the baseline methods in categories with or without graph dependency under almost all settings. For instance, without graph dependency for inference, TINED is better than baselines, except Arxiv and Pubmed where close performance is achieved. As an example, on A-computer under *ind* setting, TINED achieves 82.83% accuracy, significantly improving GLNN* by 1.73%. With graph dependency, TINED+ surpasses NOS-MOG methods on all datasets except *prod* and *tran* setting on Cora, where the performance is comparable. For instance, on Citeseer under *prod*, TINED+ achieves performance gain of 2.98% and 2.25% over NOSMOG and NOSMOG* respectively. Moreover, on Arxiv and Products with a significant distribution shift between training and test data (Zhang et al., 2022b), TINED+ outperforms the teacher and NOS-MOG, showing the capability of our techniques on large real-world graph datasets. We conclude that TINED and TINED+ can achieve excellent performance in *prod* setting with *ind* & *tran*.

### 5.4. Inference Time

Figure 3 shows the trade-off between accuracy and inference time on Citeseer. Methods closer to the top-left corner achieve a better balance of accuracy and speed. Our methods, TINED and TINED+, achieve the highest accuracy

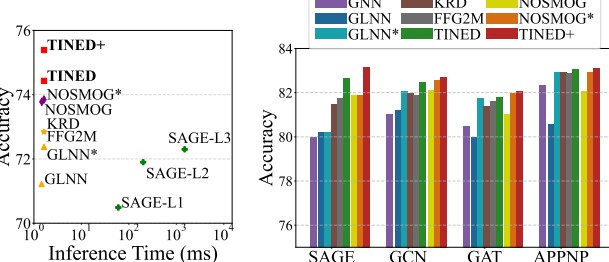

Figure 3: Inference Time and Accuracy

Figure 4: Accuracy on Different Teacher GNNs

(74.43% and 75.39%) while being fast (1.63ms and 1.64ms). In contrast, GNNs are slower, e.g., 2-layer GraphSAGE (SAGE-L2) takes 153.14ms, and 3-layer GraphSAGE (SAGE-L3) takes 1202.45ms. TINED and TINED+ are 94 times faster than SAGE-L2 and 733 times faster than SAGE-L3. All the distillation methods (NOSMOG, GLNN*, TINED, TINED+, KRD and FFG2M) have similar efficiency in around 1-2ms, with negligible differences. Compared with SAGE teacher that needs 153.14ms, the efficiency of distillation techniques in our design is validated.

### 5.5. Different Teacher GNN Architectures

Figure 4, we show that TINED and TINED+ can maintain strong performance with alternative GNN teachers, includ-

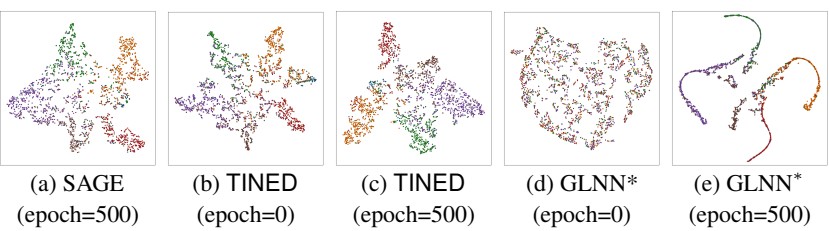

|        | (a) SAGE | (b) TINED | (c) TINED | (d) GLNN* | (e) GLNN* |
|        | (epoch=500) | (epoch=0) | (epoch=500) | (epoch=0) | (epoch=500) |

Figure 5: t-SNE of model embeddings at different training stages on Citeseer.

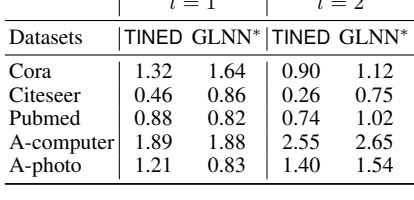

Table 3: Approximation error

| Datasets | $l=1$ TINED | $l=1$ GLNN* | $l=2$ TINED | $l=2$ GLNN* |
|---|---|---|---|---|
| Cora | 1.32 | 1.64 | 0.90 | 1.12 |
| Citeseer | 0.46 | 0.86 | 0.26 | 0.75 |
| Pubmed | 0.88 | 0.82 | 0.74 | 1.02 |
| A-computer | 1.89 | 1.88 | 2.55 | 2.65 |
| A-photo | 1.21 | 0.83 | 1.40 | 1.54 |

Figure 6: Learned DE ratios in MLPs v.s. ground truth of 2-layer GraphSAGE.

Table 4: Ablation study

| Datasets | TINED-TIN | TINED-DED | TINED | $\Delta_{\text{TIN}}$ | $\Delta_{\text{DED}}$ |
|---|---|---|---|---|---|
| Cora | 81.32±1.60 | 81.64±1.64 | 82.63±1.57 | ↑ 1.31% | ↑ 0.99% |
| Citeseer | 71.88±1.27 | 74.12±1.42 | 74.43±1.53 | ↑ 2.55% | ↑ 0.31% |
| Pubmed | 75.93±3.40 | 76.45±2.68 | 77.09±2.14 | ↑ 1.16% | ↑ 0.64% |
| A-computer | 84.59±1.78 | 84.80±1.67 | 85.18±1.12 | ↑ 0.59% | ↑ 0.38% |
| A-photo | 93.90±0.52 | 93.67±0.69 | 93.97±0.53 | ↑ 0.07% | ↑ 0.30% |
| Arxiv | 63.72±0.89 | 64.39±0.80 | 64.44±0.72 | ↑ 0.72% | ↑ 0.05% |
| Products | 69.20±0.25 | 68.63±0.26 | 69.48±0.25 | ↑ 0.28% | ↑ 0.85% |

ing GCN, GAT, and APPNP, in addition to GraphSAGE. We report the average performance of a distillation method with different teacher GNNs across the five benchmark datasets. With different teachers, TINED can always learn more effective student MLPs than GLNN, GLNN*, KRD, and FFG2M. With graph dependency, TINED+ can maintain its strong performance with different teachers and surpasses NOSMOG and NOSMOG*. When different teacher GNNs are adopted, Figure 4 illustrates the effectiveness of our methods to preserve layer-level knowledge into MLPs.

### 5.6. Experimental Analysis

**Ablation Study.** Denote TINED-TIN as TINED without Teacher Injection (TIN), and TINED-DED as TINED without Dirichlet Energy Distillation (DED).The results are in Table 4. Both TIN and DED contribute positively, and TINED achieves the best performance on all datasets, demonstrating the effectiveness of our techniques in TINED.

**Visualization.** Based on the visualization of teacher SAGE's output embeddings after 500 epochs of training (Figure 5(a)), we compare the embeddings of TINED and GLNN* both at initialization (epoch 0) and after training convergence (epoch 500). Figure 5(b) shows TINED initialized (epoch 0) using the proposed teacher injection technique, resulting in an embedding visualization similar to that of the teacher in Figure 5(a). In contrast, the visualization of GLNN* at epoch 0 appears quite random, as seen in Figure 5(d). Upon reaching training convergence (epoch 500), TINED in Figure 5(c) maintains a visualization closely resembling the teacher's in Figure 5(a), whereas the baseline GLNN* displays a noticeably different visualization. This evaluation confirms the effectiveness of the proposed teacher injection and Dirichlet energy distillation techniques employed by TINED. More visualization results are in A.5.

Table 5: Vary $\eta$ of TINED on transductive setting. Best results are in bold.

| $\eta$ | Cora | Citeseer | Pubmed | A-computer | A-photo |
|---|---|---|---|---|---|
| 1e-09 | 73.80±2.56 | 67.20±1.86 | 73.39±2.10 | 72.53±2.17 | 86.06±2.38 |
| 1e-06 | 73.50±2.48 | 67.19±1.87 | 73.32±1.88 | 72.73±1.99 | 86.35±1.79 |
| 0.001 | 77.12±0.91 | 67.24±1.83 | 73.69±1.83 | 84.13±1.23 | 87.30±2.08 |
| 0.01 | 79.26±1.30 | 69.38±1.70 | 75.12±2.32 | **85.17±1.21** | 90.48±1.38 |
| 0.1 | 81.40±1.69 | 71.52±1.58 | 76.55±2.81 | 83.61±1.67 | **93.97±0.54** |
| 0.5 | 82.01±1.64 | **74.57±1.42** | **77.10±2.15** | 83.23±1.29 | 93.51±0.60 |
| 1.0 | **82.61±1.58** | 73.57±1.39 | 76.65±2.77 | 82.85±0.91 | 93.37±0.51 |
| 10.0 | 78.85±1.63 | 73.56±1.48 | 75.61±2.73 | 70.63±4.87 | 89.53±0.78 |

**DE ratio.** Recall that Figure 1 shows the ground-truth DE ratios of teacher GraphSAGE, and we propose Dirichlet Energy Distillation in Section 4.2 to preserve smoothing effects into MLPs. Figure 6 reports the learned DE ratios of TINED, TINED-DED, and GLNN*. TINED is more closely aligned to the ground-truth DE ratios, highlighting that TINED effectively retains the GNN smoothing properties into MLPs.

**Approximation Bound.** Theorem 4.1 shows a bound between $\text{GP}^{(l)}$ in GNNs and its counterpart $\text{FC}^{l,1}$ in MLPs. Table 3 reports the errors of TINED and GLNN* using $\text{FC}^{l,1}$ to approximate $\text{GP}^{(l)}$, $||\text{GP}^{(l)}(\mathbf{H}) - \text{FC}^{l,1}(\mathbf{H})||_F/||\mathbf{H}||_F$, for $l = 1, 2$. When $l = 2$, the errors of TINED are lower than GLNN*, while the errors for $l = 1$ are comparable, showing TINED effectively approximates $\text{GP}^{(l)}$.

**Parameter Sensitivity Analysis.** In our method, the parameter $\eta$ controls the degree of fine-tuning in Eq. 6, while $\beta$ controls the importance of Dirichlet Energy Distillation in Eq. 9. Tables 5 and 7 present the accuracy results of varying $\eta$ and $\beta$ of TINED across multiple datasets. In Table 5, as $\eta$ increases, TINED demonstrates a clear trend where performance initially improves and then declines, with the best results highlighted in bold. A similar pattern is observed for

Table 6: Experiment result under *prod* (*ind* & *tran*) setting on heterophilic datasets. The best result in each category is in bold.

| Datasets | Eval | Teacher | | Without Graph Dependency | | | | | With Graph Dependency | | |
|---|---|---|---|---|---|---|---|---|---|---|---|
| | | SAGE | | FFG2M | KRD | GLNN | GLNN* | TINED | | NOSMOG | NOSMOG* | TINED+ |
| Squirrel | *prod* | 35.47 | | 38.34 | 37.16 | 39.90 | 39.70 | **41.95** | | 38.17 | 39.43 | **40.89** |
| | *ind* | 41.44±4.66 | | 42.00±4.78 | 42.11±4.77 | 44.89±5.67 | 45.00±4.56 | **46.89±5.23** | | 45.44±4.75 | 44.33±3.24 | **46.78±3.80** |
| | *tran* | 33.98±1.66 | | 37.43±3.34 | 35.93±3.36 | 38.65±1.09 | 38.37±1.02 | **40.72±1.36** | | 36.35±1.69 | 38.20±1.59 | **39.42±1.05** |
| Amazon-ratings | *prod* | 47.55 | | 50.33 | 49.56 | 49.87 | 49.41 | **50.70** | | 47.86 | 48.80 | **50.42** |
| | *ind* | 47.45±1.48 | | 47.55±0.97 | 47.68±1.00 | 47.72±1.00 | 47.71±1.14 | **49.02±1.02** | | 47.47±1.45 | 48.46±1.37 | **49.51±1.54** |
| | *tran* | 47.58±0.48 | | 51.03±1.73 | 50.03±1.50 | 50.41±0.45 | 49.84±0.39 | **51.12±0.44** | | 47.96±0.31 | 48.88±0.56 | **50.65±0.56** |

Table 7: Vary $\beta$ of TINED on transductive setting. Best results are in bold.

| $\beta$ | Cora | Citeseer | Pubmed | A-computer | A-photo |
|---|---|---|---|---|---|
| 1e-09 | 81.33±1.49 | 73.39±1.28 | 76.63±2.42 | 84.70±1.15 | 93.41±0.61 |
| 1e-06 | 81.64±1.71 | 73.39±1.30 | 76.28±2.76 | **85.17±1.21** | 93.48±0.65 |
| 0.001 | 81.64±1.57 | 73.39±1.31 | 76.51±2.64 | 84.80±1.21 | 93.65±0.68 |
| 0.1 | 81.71±1.59 | 73.65±1.38 | **77.10±2.15** | 84.70±0.99 | **93.97±0.58** |
| 1.0 | **82.61±1.58** | 73.81±1.29 | 75.84±2.66 | 84.29±1.08 | 86.59±5.66 |
| 10.0 | 80.92±2.15 | **74.57±1.42** | 71.57±2.76 | 71.35±6.30 | 78.00±3.19 |

Table 8: Compare with MLP* with 4 layers under transductive setting.

| | Cora | Citeseer | Pubmed | A-computer | A-photo |
|---|---|---|---|---|---|
| MLP | 60.84±1.08 | 63.41±1.96 | 69.41±2.88 | 70.07±1.77 | 80.19±1.48 |
| MLP* | 58.20±3.16 | 60.81±3.48 | 67.61±1.51 | 66.84±3.03 | 78.66±2.48 |
| TINED | 82.63±1.57 | 74.43±1.53 | 77.09±2.14 | 85.18±1.12 | 93.97±0.53 |

Table 9: Comparison with MLP* with 4 layers under *prod*(*ind*&*tran*) setting

| | Eval | Cora | Citeseer | Pubmed | A-computer | A-photo |
|---|---|---|---|---|---|---|
| MLP | *ind* | 61.31±2.16 | 63.95±2.95 | 69.66±2.68 | 70.36±2.48 | 79.76±2.00 |
| | *tran* | 60.88±1.41 | 62.99±2.39 | 69.67±2.61 | 69.92±2.03 | 79.53±2.05 |
| MLP* | *ind* | 58.67±2.42 | 62.15±3.50 | 67.76±1.88 | 68.09±2.60 | 77.27±2.17 |
| | *tran* | 58.12±1.81 | 61.46±2.46 | 68.07±1.87 | 67.90±2.41 | 77.05±2.48 |
| TINED | *ind* | 74.38±1.28 | 72.68±1.97 | 75.64±3.02 | 82.83±1.45 | 91.96±0.72 |
| | *tran* | 80.04±1.50 | 72.20±1.66 | 75.83±2.81 | 84.87±1.38 | 93.74±0.51 |

$\beta$ in Table 7, emphasizing the trade-off controlled by these parameters. Note that hyperparameter tuning is essential in machine learning research. Appendix A.7 of the paper details the search space for our parameters.

**Heterophilic Datasets** We conducted experiments on representative heterophilic datasets, Squirrel and Amazon-ratings, under the production setting. Squirrel (Rozemberczki et al., 2021) is a web page dataset collected from Wikipedia, while Amazon-ratings (Platonov et al., 2023) is a product co-purchasing network based on data from SNAP Datasets. In Table 6, our methods, TINED and TINED+, consistently outperform existing approaches across all settings, often by a significant margin. For instance, on the Squirrel dataset without graph dependency, TINED achieves a *prod* performance of 41.95%, surpassing the best competitor GLNN, which achieves 39.90%. Similarly, on the Amazon-ratings dataset with graph dependency, TINED+ achieves a *prod* accuracy of 50.42%, representing a 1.62% improvement over the best competitor NOSMOG*. These results highlight the effectiveness of our proposed methods in handling heterophilic datasets.

**Comparison with Vanilla MLP with Additional Layers.** In addition to the vanilla MLP with 2 layers, we also evaluate a 4-layer MLP, denoted as MLP*. Tables 8 and 9 present the results under both transductive and production settings. Our method, TINED, consistently outperforms MLP* across all datasets. Interestingly, MLP* demonstrates degraded performance compared to the 2-layer MLP, suggesting that additional layers lead to overfitting on these datasets. For instance, on the Pubmed dataset, MLP* achieves 67.61%, which is inferior to the 69.41% achieved by the 2-layer

MLP. In contrast, TINED significantly improves upon MLP*, achieving a notable accuracy of 77.09%, highlighting its robustness and effectiveness.

## 6. Conclusion

We introduce TINED, a novel method for distilling GNN knowledge into MLPs. Our approach includes Teacher Injection, which directly transfers well-trained parameters from GNNs to MLPs, and Dirichlet Energy Distillation, which preserves the unique smoothing effects of key GNN operations within MLPs. Comprehensive experiments demonstrate that TINED outperforms existing methods across various settings and seven datasets. Currently, the number of layers in the student MLP is dependent on the number of layers in the GNN teacher. In future work, we aim to develop new techniques to accelerate the process and allow for a flexible number of fully connected layers by considering the intrinsic properties of GNNs and MLPs. Moreover, we will explore the potential of TINED in other types of grpahs, including heterogeneous graphs and dynamic graphs, to further enhance its applicability and effectiveness.

## Acknowledgements

This work is supported by grants from the Research Grants Council of Hong Kong Special Administrative Region, China (No. PolyU 25201221, No. PolyU 15205224), and NSFC No. 62202404. This project has been supported by the Hong Kong Research Grants Council under General Research Fund (project no. 15200023) as well as Research Impact Fund (project no. R1015-23). This work is supported by Otto Poon Charitable Foundation Smart Cities Research Institute (SCRI) P0051036-P0050643, and grant P0048511 from Tencent Technology Co., Ltd.

## Impact Statement

This paper focuses on GNNs-to-MLPs distillation, which is a fundamental problem. There are many potential societal consequences of our work, none of which we feel must be specifically highlighted here.

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

# A. Appendix □

## A.1. Datasets

In Table 10, we provide the statistics of the datasets in experiments. For all datasets, we follow the setting in (Zhang et al., 2022b; Yang et al., 2021) to split the data. Specifically, for the first five datasets, we use the splitting in (Yang et al., 2021) and each random seed corresponds to a different split. For the OGB datasets Arxiv and Products, we follow the OGB official splits based on time and popularity respectively.

## A.2. Proof of Theorem 4.1

*Proof.* We denote the $\mathbf{H}^{\dagger}$ as the pseudo-inverse of matrix $\mathbf{H}$. Since $\mathbf{rank}(\mathbf{H}) = d$, $\mathbf{H}$ has linearly independent columns, its pseudo-inverse could be represented as $\mathbf{H}^{\dagger} = (\mathbf{H}^{\top}\mathbf{H})^{-1}\mathbf{H}^{\top}$. Moreover, given the SVD factorization of matrix $\mathbf{H} = \mathbf{U} \begin{pmatrix} \mathbf{S} \\ 0 \end{pmatrix} \mathbf{V}^{\top}$, where $\mathbf{U} \in \mathbb{R}^{N \times N}$ and $\mathbf{V} \in \mathbb{R}^{d \times d}$ are unitary, and $\mathbf{S} \in \mathbb{R}^{d \times d}$ is invertible diagonal matrix, we have $\mathbf{H}^{\dagger} = \mathbf{V} \begin{pmatrix} \mathbf{S}^{-1} & 0 \end{pmatrix} \mathbf{U}^{\top}$.

Denote $\bar{\mathbf{H}} = \mathbf{LH}$, then we have

$$||\bar{\mathbf{H}} - \mathbf{HW}||_F^2 = \sum_{i=1}^{d} ||\bar{\mathbf{H}}_{:,i} - \mathbf{HW}_{:,i}||_2^2 \quad (10)$$

Let $\mathbf{W}^* = \mathbf{H}^{\dagger}\bar{\mathbf{H}}$, we have $\mathbf{W}^*_{:,i} = \mathbf{H}^{\dagger}\bar{\mathbf{H}}_{:,i}$. Then

$$\begin{aligned} \mathbf{HW}^*_{:,i} &= \mathbf{HH}^{\dagger}\bar{\mathbf{H}}_{:,i} \\ &= \mathbf{U} \begin{pmatrix} I_d & 0 \\ 0 & 0_{n-d} \end{pmatrix} \mathbf{U}^{\top}\bar{\mathbf{H}}_{:,i} \end{aligned} \quad (11)$$

Then

$$\begin{aligned} ||\bar{\mathbf{H}}_{:,i} - \mathbf{HW}^*_{:,i}||_2 &= ||\bar{\mathbf{H}}_{:,i} - \mathbf{HH}^{\dagger}\bar{\mathbf{H}}_{:,i}||_2 \\ &\leq ||I_n - \mathbf{HH}^{\dagger}||_2 ||\bar{\mathbf{H}}_{:,i}||_2 \\ &\leq ||\bar{\mathbf{H}}_{:,i}||_2 \\ &= ||\mathbf{LH}_{:,i}||_2 \\ &\leq ||\mathbf{L}||_2 ||\mathbf{H}_{:,i}||_2 \\ &= \lambda_{\max}(\mathbf{L}) ||\mathbf{H}_{:,i}||_2 \end{aligned} \quad (12)$$

Thus we have

$$\begin{aligned} ||\bar{\mathbf{H}} - \mathbf{HW}||_F^2 &= \sum_{i=1}^{d} ||\bar{\mathbf{H}}_{:,i} - \mathbf{HW}_{:,i}||_2^2 \\ &\leq \lambda_{\max}^2(\mathbf{L}) \sum_{i=1}^{d} ||\mathbf{H}_{:,i}||_2^2 \\ &= \lambda_{\max}^2(\mathbf{L}) ||\mathbf{H}||_F^2. \end{aligned} \quad (13)$$

Finally we get

$$\frac{||\mathbf{LH} - \mathbf{HW}^*||_F}{||\mathbf{H}||_F} \leq \lambda_{\max}(\mathbf{L}) \quad (14)$$

## A.3. Implementing Other GNNs as Teacher

Here we explain how to implement other GNNs as teacher in TINED, especially on how to apply teacher injection in The Proposed Model section to different GNNs.

**GCN.** The message passing of GCN in the $l$-th layer is

$$\begin{aligned} \text{GP}^{(l)}: \tilde{\mathbf{h}}_v^{(l)} &= \hat{\mathbf{L}}\mathbf{H}^{(l-1)}, \\ \text{FT}^{(l)}: \mathbf{h}_v^{(l)} &= \sigma\left(\tilde{\mathbf{h}}_v^{(l)} \cdot \mathbf{W}^{(l)} + \mathbf{b}^{(l)}\right). \end{aligned} \quad (15)$$

where $\hat{\mathbf{L}} = \hat{\mathbf{D}}^{-\frac{1}{2}}\hat{\mathbf{A}}\hat{\mathbf{D}}^{-\frac{1}{2}}$ is the normalized Laplacian of the graph, $\hat{\mathbf{A}} = \mathbf{A} + I_n$ is the adjacency matrix with self loops, $\hat{\mathbf{D}}_{ii} = \sum_j \hat{\mathbf{A}}_{ij}$ is the degree matrix, $\mathbf{W}^{(l)}$ is the transformation matrix of $l$-th layer, and $\mathbf{H}^{(l)}$ is the output of $l$-th layer.

For the $l$-th layer of the GCN teacher, we directly inject $\mathbf{W}^{(l)}$ and $\mathbf{b}^{(l)}$ of the teacher in Equation (15) into the student FC layer $\text{FC}^{l,2}$ in Equation (5), i.e., $\mathbf{W}_M^{l,2} = \mathbf{W}^{(l)}$ and $\mathbf{b}_M^{l,2} = \mathbf{b}^{(l)}$, while using $\text{FC}^{l,1}$ in Equation (5) to simulate $\text{GP}^{(l)}$ in Equation (15).

**GAT.** The message passing of GAT in the $l$-th layer with single head attention is

$$\begin{aligned} \text{GP}^{(l)}: \tilde{\mathbf{h}}_v^{(l)} &= \Pi^{(l)}\mathbf{H}^{(l-1)}, \\ \text{FT}^{(l)}: \mathbf{h}_v^{(l)} &= \sigma\left(\tilde{\mathbf{h}}_v^{(l)} \cdot \mathbf{W}^{(l)} + \mathbf{b}^{(l)}\right). \end{aligned} \quad (16)$$

where $\Pi^{(l)}$ is the attention matrix defined as $\Pi_{ij}^{(l)} = \text{LeakyReLU}([\mathbf{H}_i^{(l-1)}\mathbf{W}^{(l)}||\mathbf{H}_j^{(l-1)}\mathbf{W}^{(l)}]^{\top}\mathbf{a}^{(l)})$ if $(i,j)$ is an edge in $\mathcal{G}$, i.e., $(i,j) \in \mathcal{E}$, and $\Pi_{ij}^{(l)} = 0$ otherwise, $||$ denotes the concatenation operation, and $\mathbf{a}^{(l)} \in \mathbb{R}^{2d_l}$ is a learnable attention vector.

For the $l$-th layer of the GAT teacher, we directly inject $\mathbf{W}^{(l)}$ and $\mathbf{b}^{(l)}$ of the teacher in Equation (16) into the student, i.e., $\mathbf{W}_M^{l,2} = \mathbf{W}^{(l)}$ and $\mathbf{b}_M^{l,2} = \mathbf{b}^{(l)}$, while using $\text{FC}^{l,1}$ in Equation (5) to simulate $\text{GP}^{(l)}$ in Equation (16).

**APPNP.** APPNP decouples feature transformation and graph propagation into two stages. APPNP first performs FTs for $T_1$ times in Equation (17) to get $\mathbf{H}^{(T_1)}$ (usually $T_1 = 2$), and then propagates $\mathbf{H}^{(T_1)}$ over the graph for $T_2$ hops in Equation (18) by approximate Personalized PageRank.

$$\text{FT}^{(l)}: \mathbf{H}^{(l)} = \mathbf{H}^{(l-1)}\mathbf{W}^{(l)}, l = 1, 2, ..., T_1, \quad (17)$$

$$\text{GP}: \mathbf{H}^{(l)} = (1-\alpha)\hat{\mathbf{L}}\mathbf{H}^{(l)} + \alpha\mathbf{H}^{(T_1)}, l = T_1+1, T_1+2, ..., T_1+T_2, \quad (18)$$

where $\alpha$ is a hyperparameter and $\hat{\mathbf{L}}$ is the normalized Laplacian mentioned before.

Table 10: Dataset Statistics and Splits.

| Dataset | # Nodes | # Edges | # Features | # Classes | # Train | # Val | # Test |
|---------|---------|---------|------------|-----------|---------|-------|--------|
| Cora | 2,485 | 5,069 | 1,433 | 7 | 140 | 210 | 2135 |
| Citeseer | 2,110 | 3,668 | 3,703 | 6 | 120 | 180 | 1810 |
| Pubmed | 19,717 | 44,324 | 500 | 3 | 60 | 90 | 19567 |
| A-computer | 13,381 | 245,778 | 767 | 10 | 200 | 300 | 12881 |
| A-photo | 7,487 | 119,043 | 745 | 8 | 160 | 240 | 7087 |
| Arxiv | 169,343 | 1,166,243 | 128 | 40 | 90941 | 29799 | 48603 |
| Products | 2,449,029 | 61,859,140 | 100 | 47 | 196615 | 39323 | 2213091 |

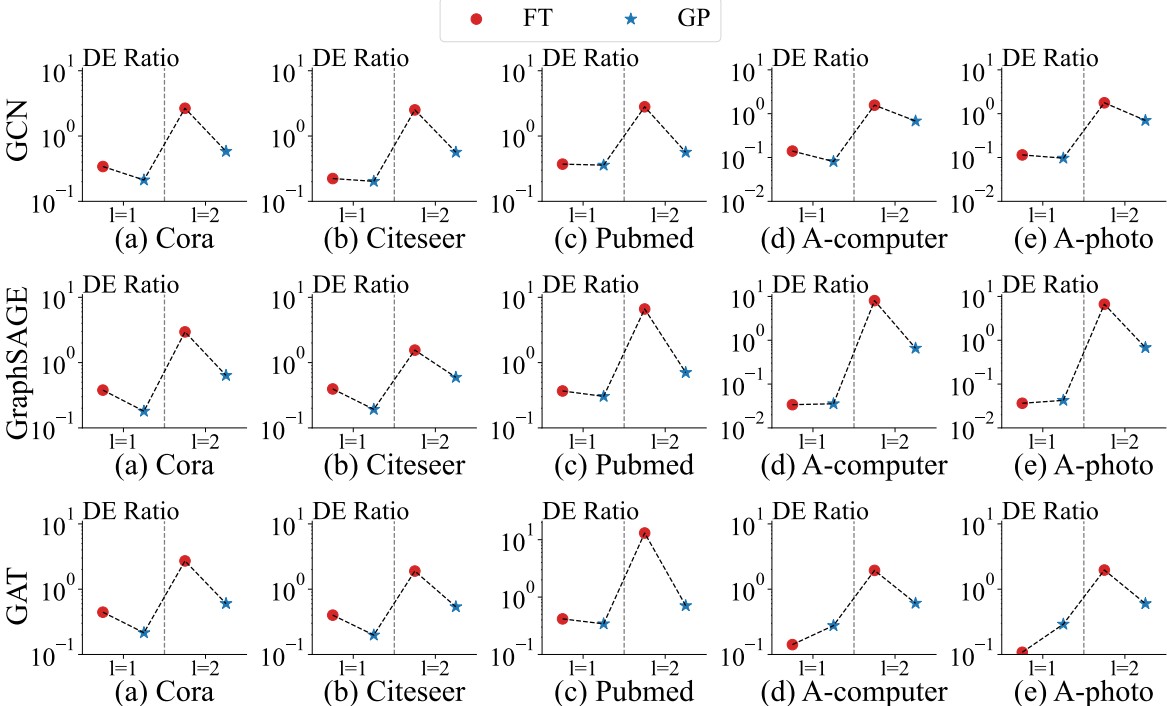

Figure 7: The layer-wise DE ratio of trained GCN, GraphSAGE and GAT, with 2 layers architecture on different datasets. The $x$-axis is the layer number, and the $y$-axis is the DE ratio value.

For each of $\mathrm{FT}^{(l)}$ in APPNP, we inject it into a $\mathrm{FC}^l$ layer in the student MLP. In other words, we have $T_1$ FC layers to mimic the FT layers in APPNP. Then for the $T_2$ steps of propagation in Equation (18), we use one FC layer $\mathrm{FC}^{T_1+1}$ to mimic all of them, since $T_2$ can be large. In total, there are $T_1 + 1$ FC layers in MLP to approximate the teacher GNN. The DED loss of APPNP is $\mathcal{L}_{DED} = \sum_{l=1}^{T_1}(R_{\mathrm{FT}^{(l)}} - R_{\mathrm{FC}^l})^2 + (R_{\mathrm{GP}} - R_{\mathrm{FC}^{T_1+1}})^2$

### A.4. DE Ratios of other GNN architectures

To show the pattern of DE ratios in different GNN teachers, we plot the layer-wise DE ratios of GCN, GraphSAGE and GAT, in addition to GraphSAGE, with 2 layers architecture on the five benchmarks in Figure 7. From Figure 7 we can observe that all GNNs share very similar patterns across datasets: within a specific layer, in Figure 7, we observe

the following nearly consistent patterns about *DE-ratio*: (i) within the same layer for $l = 1, 2$, most DE ratio $R_{\mathrm{FT}^{(l)}}$ for $\mathrm{FT}^{(l)}$ is larger than $R_{\mathrm{GP}^{(l)}}$ of $\mathrm{GP}^{(l)}$, suggesting that $\mathrm{GP}^{(l)}$ actively smooths embeddings, whereas $\mathrm{FT}^{(l)}$ is relatively conservative for smoothing; (ii) at $l = 2$, DE ratio $R_{\mathrm{FT}^{(l)}}$ even surpasses 1, indicating that in this layer, $\mathrm{FT}^{(l)}$ acts to diversify embeddings rather than smoothing them. Furthermore, to investigate the smoothing behavior on deeper layer cases, we plot Figure 8 showing layer-wise DE ratios of 5-layer trained GNNs, and from Figure 8 we can observe that the trained 5-layer GNNs have similar patterns to the 2-layer GNNs: the DE ratios of $\mathrm{FT}^{(l)}$ is generally higher than that of $\mathrm{GP}^{(l)}$, indicating that $\mathrm{GP}^{(l)}$ is aggressive while $\mathrm{FT}^{(l)}$ is relatively conservative for smoothing.

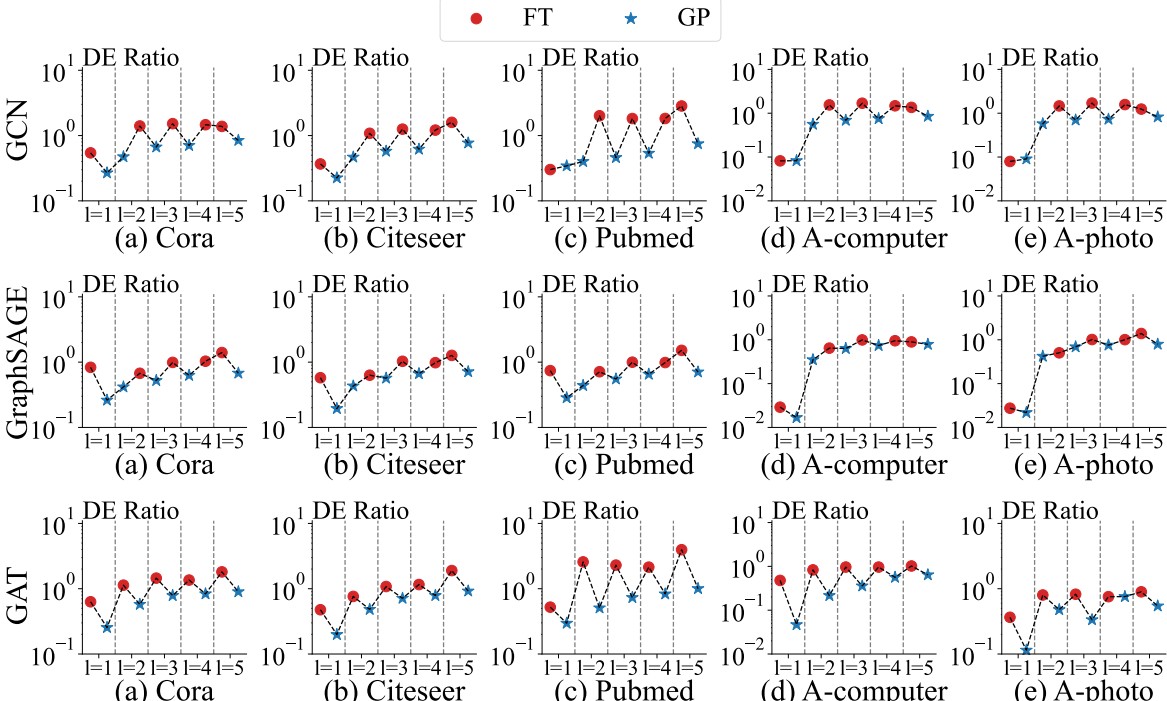

Figure 8: The layer-wise DE ratio of trained GCN, GraphSAGE and GAT, with 5 layers architecture on different datasets. The $x$-axis is the layer number, and the $y$-axis is the DE ratio value.

## A.5. Layer-wise t-SNE Visualization

In Figure 9, we present a detailed layer-wise t-SNE visualization of the embeddings for various models and stages. Figure 9(a) (1st row) shows the 2-layer GraphSAGE teacher model. Figure 9(b) (2nd row) illustrates TINED at initialization before training (epoch 0), while Figure 9(c) (3rd row) displays TINED after training convergence (epoch 500). Figure 9(d) (4th row) and Figure 9(e) (5th row) depict the baseline GLNN* at initialization (epoch 0) and after training convergence (epoch 500), respectively. For the 2-layer GraphSAGE in Figure 9(a), the 1st row visualizes the embeddings from left to right: after the $\mathrm{FT}^{(1)}$ operation in the first layer, the $\mathrm{GP}^{(1)}$ operation in the first layer, $\mathrm{FT}^{(2)}$ in the second layer, and $\mathrm{GP}^{(2)}$ in the second layer. For each distillation method in Figures 9(b-e), each row contains four plots representing the embeddings of the four fully-connected (FC) layers in the methods, corresponding to the four operations in the first row of the teacher. Compared to the teacher model in Figure 9(a), TINED at initialization in Figure 9(b) already exhibits quite similar embedding patterns, thanks to the proposed teacher injection technique. In contrast, the baseline GLNN* shows a random visual pattern at epoch 0 in Figure 9(d), which is radically different from the teacher model. After training convergence at 500 epochs, TINED's visualized embeddings in Figure 9(c) closely resemble those of the teacher in Figure 9(a), whereas

the baseline GLNN* in Figure 9(e) produces embeddings that are visually distinct. These observations demonstrate the effectiveness of the teacher injection and Dirichlet energy distillation techniques in TINED.

## A.6. Computer Resource Details

The experiments on both baselines and our approach are implemented using PyTorch, the DGL (Wang et al., 2019) library for GNN algorithms, and Adam (Kingma and Ba, 2015) for optimization. We run all experiments on Intel(R) Xeon(R) Platinum 8338C CPU @ 2.60GHz CPU, and a Nvidia Geforce 3090 Cards with Cuda version 11.7. Source codes of all competitors are obtained from respective authors. The totally training time cost for one set of hyper-parameters varies among datasets, from Cora costing 10 minutes to Products costing 13 hours.

## A.7. Teacher details and hyperparameter search space

The hyperparameters of GNN models on each dataset are taken from the best hyperparameters provided by previous studies (Tian et al., 2022; Zhang et al., 2022b). For APPNP teacher model, we found that the suggested hyperparameter in previous literature (Zhang et al., 2022b; Tian et al., 2022) produces relatively poor results, thus we search it in following space, achieving a better APPNP teacher: # layers from $[2, 3]$, learning rate from $[0.0001, 0.001, 0.01]$,

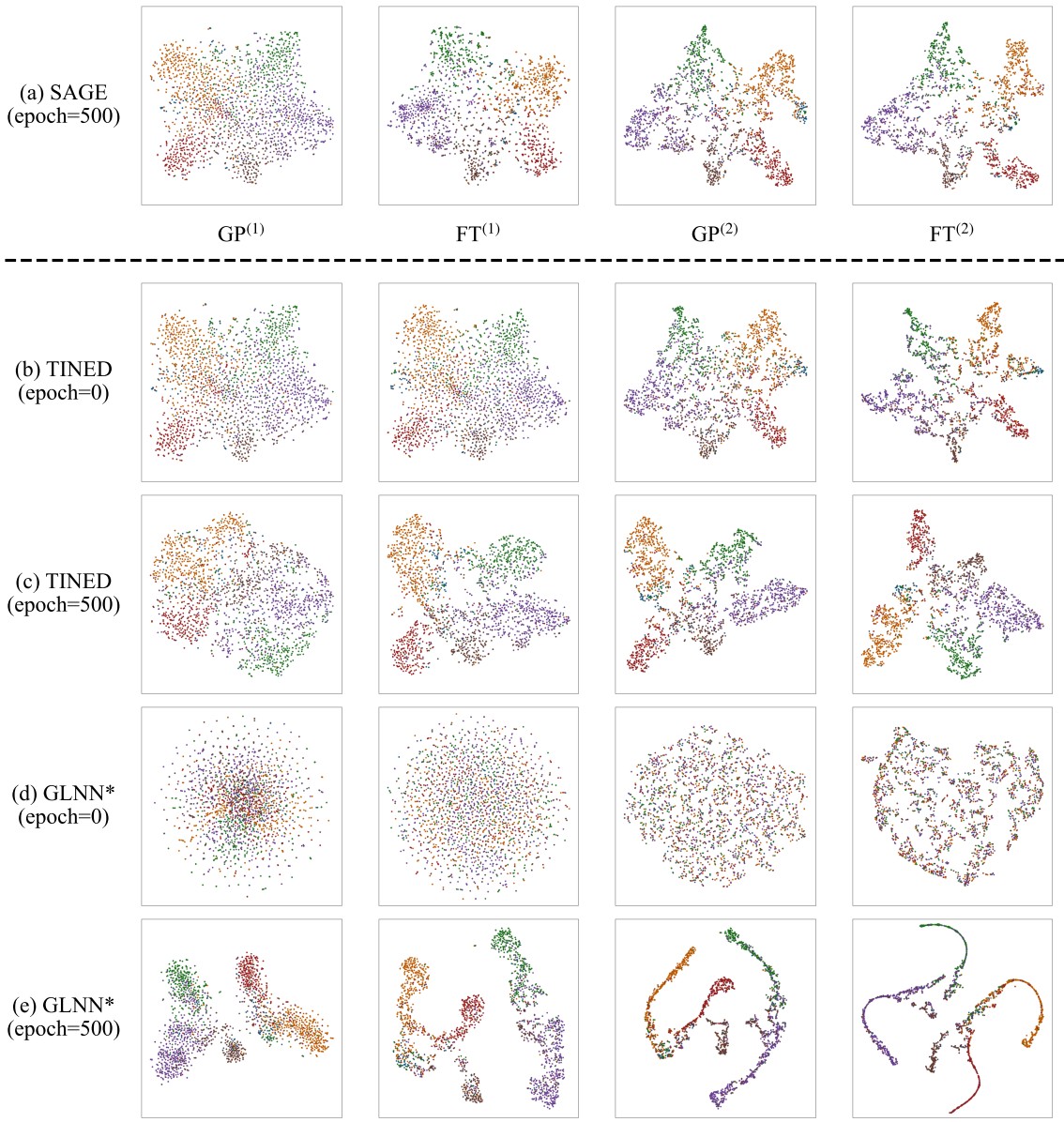

Figure 9: Layer-wise t-SNE visualization of the embeddings for various models (GraphSAGE, TINED, GLNN*) and stages (epoch 0 at initialization and epoch 500 after training convergence) on Citeseer.

weight decay from $[0.0001, 0.001, 0.01]$, dropout from $[0, 0.5, 0.8]$, hidden dim from $[128, 256]$, power iteration $K$ from $[5, 10, 15]$. For the students MLP, GLNN and NOSMOG, we set the number of layers and the hidden dimension of each layer to be the same as the teacher GNN, so their total number of parameters stays the same as the teacher GNN. For TINED, TINED+, GLNN* and NOSMOG*, given a layer-wise GNN teacher with $T$ layers (including, GraphSAGE, GCN, and GAT), in the student MLP we set the same hidden dimension as teachers, and $2T$ layers, while for APPNP, we set the student MLP to have $T_1 + 1$ layers as explained in Appendix A.3. Other

hyper-parameter searching spaces are listed here: Learning rate from $[0.0001, 0.0005, 0.001, 0.005, 0.01]$, weight decay from $[0.0, 0.0001, 0.0005, 0.001, 0.005, 0.01]$, weight of distillation $\lambda$ from $[0.1, 0.4, 0.5, 0.6, 1]$, nornamlization type from [batch normalization, layer normalization, none], dropout from $[0, 0.1, 0.3, 0.5, 0.8]$. Batch size for two large OGB datasets from $[512, 1024, 4096]$. Weight of DED $\beta$ from $[1e^{-6}, 5e^{-5}, 1e^{-5}, 0.05, 0.1, 0.5, 1, 5, 10]$. Fine tuning weight $\eta$ for injected teacher FT layers from $[0.01, 0.1, 0.5, 1, 3, 10]$, For the hyperparamrter space of TINED+ from NOSMOG, the search space is the same as (Tian et al., 2022). In large OGB datasets, the direct com-

Table 11: Hyperparameters for GNNs on five datasets from (Zhang et al., 2022b; Tian et al., 2022).

| Dataset | GraphSAGE | GCN | GAT |
|---|---|---|---|
| # layers | 2 | 2 | 2 |
| hidden dim | 128 | 64 | 64 |
| learning rate | 0.01 | 0.01 | 0.01 |
| weight decay | 0.0005 | 0.001 | 0.01 |
| dropout | 0 | 0.8 | 0.6 |
| fan out | 5,5 | - | - |
| attention heads | - | - | 8 |

Table 12: Vary $\zeta$ for TINED on large graph ogbn-arxiv under transductive setting.

| $\zeta$ | 0.1 | 0.3 | 0.5 | 0.7 | >0.8 |
|---|---|---|---|---|---|
| Arxiv | 64.31±0.71 | 64.32±0.91 | 64.38±0.75 | 64.44±0.80 | OOM |

putation of DE values will run out of GPU memory 24GB. Thus, we propose a sampling ratio $\zeta$ to compute approximate DE values in DED process. This is done by inducing a subgraph based on random sampling of edges, then approximate the DE ratio of input and output feature matrix on this sampled subgraph. We also search this sampling proportion $\zeta$ from $[0.001, 0.005, 0.1, 0.4, 1]$. Moreover, empirically we found that applying smoothing term on the DE ratio in MSE loss could benefit optimization, thus we propose a smoothing function technique $\mu$ on DE ratio $R_{op}$, i.e. $\mu(R_{op})$, when computing the DED loss, where $\mu$ is searched from $[\text{sqrt}(\cdot), \log(\cdot), \text{Identity}(\cdot)]$ (Jadon et al., 2022).

### A.8. Impact of $\zeta$ on Large Datasets

The parameter $\zeta$ controls the subgraph sampling ratio used to estimate the DE ratio, helping to avoid memory overflow on large datasets. We vary $\zeta$ and report the results on the large ogbn-arxiv dataset in Table 12. The results show that TINED maintains stable performance across different $\zeta$ values. Notably, $\zeta$ values larger than 0.8 are unnecessary, as they increase computational overhead without improving distillation quality.

### A.9. Comparison with VQGraph

In this section, we include a comparison with VQGraph (Yang et al., 2024) with GraphSAGE teacher. We thoroughly searched all the hyperparameter spaces specified in the original VQGraph paper (Table 12 of its paper). Specifically, VQGraph has the following hyperparameter search space for the teacher GraphSAGE with codebook: max epoch $\in$ [100, 200, 500], hidden dim = 128, dropout ratio $\in$ [0, 0.2, 0.4, 0.6, 0.8], learning rate $\in$ [0.01, 1e-3, 1e-4], weight decay $\in$ [1e-3, 5e-4, 0], codebook size $\in$ [8192, 16384], lamb node $\in$ [0, 0.01, 0.001, 1e-4], and lamb edge $\in$ [0, 1e-1, 0.03, 0.01, 1e-3]. For distillation, the hyperpa-

Table 13: Results of VQGraph and our TINED+ with GraphSAGE as teacher on transductive setting. The best result is in bold.

| | Cora | Citeseer | Pubmed | A-computer | A-photo |
|---|---|---|---|---|---|
| VQGraph | 78.66±1.21 | 74.66±1.23 | 73.02±3.51 | 80.16±2.02 | 92.32±1.74 |
| TINED+ | **83.70±1.02** | **75.39±1.59** | **77.75±3.14** | **84.82±1.58** | **94.05±0.39** |

rameter search space of VQGraph is: max epoch $\in$ [200, 500], norm type $\in$ ["batch", "layer", "none"], hidden dim = 128, dropout ratio $\in$ [0, 0.1, 0.4, 0.5, 0.6], learning rate $\in$ [0.01, 5e-3, 3e-3, 1e-3], weight decay $\in$ [5e-3, 1e-3, 1e-4, 0], lamb soft labels $\in$ [0.5, 1], and lamb soft tokens $\in$ [1e-8, 1e-3, 1e-1, 1]. The table below reports the results of VQGraph and our method TINED+. Observe that our method outperforms VQGraph on the datasets, which validates the effectiveness of TINED.

Since VQGraph involves re-training the teacher, whereas all other competitors, including KRD, FF-G2M, GLNN, NOSMOG, and TINED, use a fixed teacher for distillation, we believe VQGraph belongs to a different category and have therefore excluded it from the main result.

