# OpenReview forum: "TINED: GNNs-to-MLPs by Teacher Injection and Dirichlet Energy Distillation"
_ICML.cc/2025/Conference — ICML 2025 poster_

### Official Review · Reviewer_cPDP · 2025-03-06

**Overall Recommendation:** 3

**Summary:**

This paper proposes a layer-wise method with Dirchlet energy imitation for node-level knowledge distillation from GNN teacher to MLP student, to reduce the latency of inference for time-sensitive applications.

**Claims And Evidence:**

The problem is well-defined, and the claims in introduction section is clear, and supported by experimental results.

**Essential References Not Discussed:**

No.

**Experimental Designs Or Analyses:**

The experimental design is detailed and comprehensive, however it lack a more recent baseline VQGraph, which has been mentioned in the related work section.

**Methods And Evaluation Criteria:**

The layer-wise distillation method is straight-forward, and make sense, but why the Dirchlet energy ratio is a suitable quantity for distillation need more justifications.

**Other Comments Or Suggestions:**

N/A

**Other Strengths And Weaknesses:**

Strengths: The experiments are comprehensive, and most recent baselines are included. This work proposes layer-wise distillation based on Dirchlet energy ratio, which quantifies the smoothness of the node embeddings.

Weaknesses:

1. The use of Dirchlet energy ratio is not well justified, i.e., why it is appropriate for knowledge trasnfer, what makes it effective to outperform previous methods?

2. The improvement compared to previous baselines are not very significant, and some baselines are missing (e.g., VQGraph)

**Questions For Authors:**

1. The biggest concern is the justification of using Dirchlet energy ratio as the quantity for knowledge transfer, and the reason why it is more effective than previous SOTA methods.

**Relation To Broader Scientific Literature:**

Compared to previous approaches, this paper proposes layer-wise distillation using energy ratio for knowledge trasnfer.

**Theoretical Claims:**

I believe the proof is correct, however, the error bound can be loose since $\lambda_{max}$ can be a large quantity especially for graphs with node degree skewed, making the theoretical result not very valuable.

---

> ### Author Rebuttal · Authors · 2025-04-01
>
> ### Thank you for appreciating the strengths of our work. Below are our responses to address your important comments.
>
>
> >**C1.** The justification of using Dirchlet energy ratio as the quantity for knowledge transfer, and the reason why it is more effective than previous SOTA methods.
>
> **Response:** Thank you for the insightful comment. Please find our justifications below. (i) The goal of distillation is to enable the student MLP to preserve the GNN teacher's knowledge as much as possible. Our analysis reveals a crucial finding: the Feature Transformation (FT) and Graph Propagation (GP) operations in a GNN layer often exhibit opposing smoothing effects. GP tends to be aggressive, while FT is more restrained in smoothing, as shown in Figure 1 of the paper. These distinct smoothing patterns represent important teacher knowledge that the student should preserve. We achieve this by leveraging the proposed Dirichlet Energy Distillation (DED), where we use the well-recognized smoothness measure, Dirichlet Energy, to design the DE ratio. This DED technique is validated to be effective in the experiments. (ii) Moreover, in our design, the parameters of FT operations in teacher GNN have been directed injected into the student by the teacher injection technique (TIN), while the GP operations are not compatible with the FC layers in student MLP, and the DED technique with Dirichlet Energy ratio helps on the distillation of GP operations. Therefore, *our methods, combining TIN and DED, work together* to achieve better performance than existing methods, as validated by extensive experimental results and ablation studies.
>
> >**C2.** I believe the proof is correct. The error bound can be loose on graphs with node degree skewed.
>
>
> **Response:** Thank you for the insightful comment. As stated in our paper, the value of our analysis lies in its attempt to establish a theoretical relationship for using simple MLP layers to approximate complex GNNs. Our theoretical analysis focuses on the general case across all possible graphs, with the error being bounded by the largest eigenvalue. We acknowledge that this bound can be large in scenarios where the node degree distribution is skewed. In experiments, the observed actual error is typically much smaller than the theoretical result, as validated in Table 3 of the paper.
>
>
>
>
>
>
>
>
> > **C3.** The experiments are comprehensive, and most recent baselines are included. Compare with a recent baseline VQGraph, which has been mentioned in the related work section.
>
> **Response:** Thank you for your feedback on the comprehensiveness of our experiments. As suggested, we have included a comparison with VQGraph. We thoroughly searched all the hyperparameter spaces specified in the original VQGraph paper (Table 12 of VQGraph). Specifically, VQGraph has the following hyperparameter search space for the teacher SAGE with codebook: max_epoch $\in$ \{100, 200, 500\}, hidden_dim $=$ 128, dropout_ratio $\in$ \{0, 0.2, 0.4, 0.6, 0.8\}, learning_rate $\in$ \{0.01, 1e-3, 1e-4\}, weight_decay $\in$ \{1e-3, 5e-4, 0\}, codebook_size $\in$ \{8192, 16384\}, lamb_node $\in$ \{0, 0.01, 0.001, 1e-4\}, and lamb_edge $\in$ \{0, 1e-1, 0.03, 0.01, 1e-3\}. For distillation, the hyperparameter search space of VQGraph is: max_epoch $\in$ \{200, 500\}, norm_type $\in$ \{“batch”, “layer”, “none”\}, hidden_dim $=$ 128, dropout_ratio $\in$ \{0, 0.1, 0.4, 0.5, 0.6\}, learning_rate $\in$ \{0.01, 5e-3, 3e-3, 1e-3\}, weight_decay $\in$ \{5e-3, 1e-3, 1e-4, 0\}, lamb_soft_labels $\in$ \{0.5, 1\}, and lamb_soft_tokens $\in$ \{1e-8, 1e-3, 1e-1, 1\}. The table below reports the results of VQGraph and our method TINED+. Observe that our method outperform VQGraph on the datasets. We will include this comparison in the paper.
>
> **Table A** Results of VQGraph and our TINED+ with SAGE as teacher on transductive setting. The best result is in italics.
> ||Cora|Citeseer|Pubmed|Computer|Photo|
> |-|-|-|-|-|-|
> |VQGraph|78.66|74.66|73.02|80.16|92.32|
> |TINED+|*83.70*|*75.39*|*77.75*|*84.82*|*94.05*|

---

### Official Review · Reviewer_JZCk · 2025-03-09

**Overall Recommendation:** 4

**Summary:**

This paper addresses the GNN2MLP distillation task, which aims to transfer knowledge from computationally expensive Graph Neural Networks (GNNs) to more efficient Multilayer Perceptrons (MLPs) for faster inference on graph-structured data.

**Claims And Evidence:**

Yes

**Essential References Not Discussed:**

N/A

**Experimental Designs Or Analyses:**

The reviewer has checked Section 5. The comparison with vanilla MLPs is unfair since TINED's MLP has more layers.

**Methods And Evaluation Criteria:**

Overall, the evaluation makes sense. But the hyperparameter search space is too broad.

**Other Comments Or Suggestions:**

There are typos in the manuscript. For example: "nornamlization" and "hyperparamrter".

**Other Strengths And Weaknesses:**

# Strengths:
This work effectively categorizes the key components of a GNN layer into two distinct parts based on their operational patterns: feature transformation and message passing. For feature transformation, which shares the same structural knowledge with student MLPs, the authors directly transfer the parameters. For message passing, which constitutes the core functionality of GNNs, the authors identify its distinct smoothing effect compared to feature transformation.

# Weaknesses:
1. The term "fine-tuning" (Equation 6) is ambiguously defined. The paper lacks clarity on how the parameter $\eta$ balances between inherited and learned parameters, which requires further elaboration.
2. This work focuses primarily on performance optimization rather than efficiency. The inference time of TINED does not show any improvement compared to other distillation baselines.

**Questions For Authors:**

Can TINED effectively handle dynamic graphs in which node features or graph structure evolve over time? If not, what modifications would be necessary to accommodate such scenarios?
The Dirichlet Energy ratio of an operation quantifies smoothing effects but may potentially conflate multiple factors, such as softmax functions and batch/layer normalization between layers. How does TINED ensure that the computed DE ratios accurately reflect the intended smoothing/diversification effects of specific operations?

**Relation To Broader Scientific Literature:**

The contributions of this work are related to GNN-to-MLP distillation and Dirichlet energy analysis.

**Theoretical Claims:**

Overall, the theoretical results look good to me.

---

> ### Author Rebuttal · Authors · 2025-04-01
>
> ### Thank you for acknowledging the strengths of our work. Below are our responses addressing your important comments.
>
> > **C1.** The hyperparameter search space.
>
> **Response:** We clarify that we mainly adopt the conventional hyperparameter search space for the teacher model and the student model, following existing studies. For the hyperparameters of our method, we also employ grid search within a limited space. Detailed information about the hyperparameters is provided in the appendix.
>
>
> > **C2.** Compare with vanilla MLP with the same number of layers as TINED.
>
> **Response:** As suggested, in addition to the vanilla MLP with 2 layers, we also compare with a 4-layer MLP, denoted as MLP*. In **Table A** and **B** below, under both transductive and production settings, our method, TINED, outperforms MLP* across all datasets. Moreover, compared to the MLP with 2 layers, MLP* shows degraded performance, indicating that additional layers cause the MLP to overfit on these datasets.
>
> **Table A: Compare with MLP\* with 4 layers under transductive setting. The results are averaged over 10 runs and reported with standard deviation**
> |   | cora | citeseer | pubmed | a-computer | a-photo |
> |-|-|-|-|-|-|
> | MLP | 60.84±1.08 | 63.41±1.96 | 69.41±2.88 | 70.07±1.77 | 80.19±1.48 |
> | MLP\* | 58.20±3.16 | 60.81±3.48 | 67.61±1.51 | 66.84±3.03 | 78.66±2.48 |
> |TINED |82.63±1.57| 74.43±1.53 | 77.09±2.14 | 85.18±1.12 | 93.97±0.53|
>
> **Table B: Compare with MLP\* with 4 layers under *prod*(*ind*&*tran*) setting**
> |   | setting | cora | citeseer | pubmed | a-computer | a-photo |
> |-|-|-|-|-|-|-|
> | MLP | *ind* | 61.31±2.16 | 63.95±2.95 | 69.66±2.68 | 70.36±2.48 | 79.76±2.00 |
> | | *tran* | 60.88±1.41 | 62.99±2.39 | 69.67±2.61 | 69.92±2.03 | 79.53±2.05 |
> | MLP\* | *ind* | 58.67±2.42 | 62.15±3.50 | 67.76±1.88 | 68.09±2.60 | 77.27±2.17 |
> | | *tran* | 58.12±1.81 | 61.46±2.46 | 68.07±1.87 | 67.90±2.41 | 77.05±2.48 |
> |TINED | *ind* | 74.38±1.28 | 72.68±1.97 | 75.64±3.02 | 82.83±1.45 | 91.96±0.72|
> | | *tran* |  80.04±1.50 | 72.20±1.66 | 75.83±2.81 | 84.87±1.38 | 93.74±0.51|
>
>
>
> > **C3.** Clarify fine-tuning with $\eta$ (Equation 6).
>
> **Response:** In Eq (6), the parameter $\eta$ controls the balance between the inherited teacher knowledge and learned parameters during fine-tuning. A small $\eta$ tends to let the student make less changes to the inherited teacher parameters, and as $\eta$ increases, the fine-tuning will update the parameters aggressively. As shown in **the experiments in the response to comment C1 of Reviewer qs85**, when varying $\eta$, a mild $\eta$ setting can strike a good balance, leading to better performance.
>
>
>
> > **C4.** This work focuses primarily on performance optimization rather than efficiency.
>
> **Response:** Our goal is to achieve better effectiveness than existing distillation methods, ensuring that the student MLP operates significantly faster than the teacher GNN, while maintaining comparable efficiency to existing methods. As shown in Figure 3 of the paper, our methods TINED and TINED+ achieve the best trade-off between accuracy and inference time. For example, on CiteSeer, while the teacher requires *153.14 milliseconds (ms)* for inference, our methods, TINED and TINED+, achieve the highest accuracy at the cost of just *1.63 ms*. All the distillation methods maintain the same order of efficiency, around *1-2 ms*.
>
>
>
> > **C5.** Can TINED effectively handle dynamic graphs? If not, what modifications would be necessary?
>
> **Response:** In this work, we focus on static GNNs, but we agree that distillation on dynamic GNNs is a promising direction. We believe that the proposed Teacher Injection (TIN) and Dirichlet Energy Distillation (DED) techniques are compatible with dynamic GNNs, such as DynGNN, EvolveGCN, and TGAT. For dynamic GNNs, TIN can be adapted to periodically inject temporal parameters from the teacher into the student, and DED can be performed adaptively since the DE ratio can be efficiently calculated on new graph snapshots. We leave a detailed investigation of these adaptations for future work.
>
>
> > **C6.**  How does TINED ensure that the computed DE ratios accurately reflect the intended smoothing/diversification effects of specific operations?
>
> **Response:** We clarify that the operation considered by the DE ratio in Definition 4.3 is the GP operation in Eq. (4), which performs aggregation and concatenation, excluding subsequent softmax and batch normalization. Our formulation precisely controls the computation boundary to prevent conflation with other factors, ensuring that the DE ratio accurately reflects only the smoothing/diversification effect of the GP operation itself. DE is a widely adopted measure in the literature for assessing the smoothing effect of GNNs [1], making it a reasonable choice for designing our DE ratio.
>
> [1] T Konstantin Rusch, Michael M Bronstein, and Siddhartha Mishra. A survey on oversmoothing in graph neural networks. 2023.

---

### Official Review · Reviewer_bbbm · 2025-03-12

**Overall Recommendation:** 4

**Summary:**

This paper introduces TINED, a method for distilling Graph Neural Networks (GNNs) into Multi-Layer Perceptrons (MLPs) via layer-wise Teacher Injection and Dirichlet Energy Distillation (DED). The key idea is to directly inject parameters from GNN feature transformation layers into MLP layers and use DED to preserve opposing smoothing effects of GNN operations, specifically feature transformation (FT) and graph propagation (GP). Theoretical bounds on approximating GP with MLP layers are also provided. Experiments conducted on seven datasets demonstrate that TINED outperforms both original GNN models and prior distillation methods.

**Claims And Evidence:**

Yes, the claims made in the submission are supported by clear and convincing evidence
The authors present the following key claims:

1. "The valuable knowledge of a teacher GNN is preserved in the well-trained parameters of its FT and GP operations." (line 50-53). This is substantiated by the ablation study (Table 6) and main benchmark results (Tables 1 and 2).
2. "The FT and GP operations in a GNN layer often exert opposing smoothing effects: GP aggressively smooths node embeddings, while FT is more restrained and can even diversify embeddings." (line 84-87). This is validated through DE ratio analysis (Figures 1 and 7).

**Essential References Not Discussed:**

The research by Winter et al. [1] examines the underlying mechanisms contributing to the effectiveness of distillation methods. Including this work in the discussion would provide additional context and insights.

[1] Winter, D., Cohen, N., & Hoshen, Y. (2024). Classifying nodes in graphs without GNNs. arXiv preprint arXiv:2402.05934.

**Experimental Designs Or Analyses:**

Yes, I have checked the soundness of the experimental designs and analyses.
The comprehensive evaluation framework incorporates both transductive and inductive/production settings (Tables 1 and 2).
The authors present a thorough set of experimental results, including: ablation studies (Table 6), TSNE visualization of learned embeddings (Figure 5), approximation errors of MLPs (Table 3), inference time comparisons (Figure 3), learned DE ratio analysis (Figure 6), parameter sensitivity analysis (Tables 4 and 5), and performance evaluations across different teacher models (Figure 4).

**Methods And Evaluation Criteria:**

Yes, the proposed methods and evaluation criteria make sense for the problem at hand.
The authors develop their methodology based on a systematic analysis of the GNN2MLP distillation problem. Their approach effectively addresses the identified challenges: preserving knowledge in teacher layers and managing the disparity between the FT and GP operations.
The evaluation employs widely recognized benchmarks to assess the performance of the proposed method.
The authors distinguish between two inference settings: graph-dependent and graph-free. This distinction corresponds to two practical scenarios: warm start and cold start, respectively. In cold start scenarios (where new nodes have no edges), graph information is inaccessible during inference. Conversely, in warm start scenarios (where new test nodes have edges), limited graph information can be utilized during inference, as demonstrated by [1]. While the authors provide a brief discussion of this distinction (line 185-190), a more comprehensive elaboration would be beneficial.

[1] Tian, Y., Zhang, C., Guo, Z., Zhang, X., & Chawla, N. (2023). Learning MLPs on graphs: A unified view of effectiveness, robustness, and efficiency. In International Conference on Learning Representations.

**Other Comments Or Suggestions:**

N.A.

**Other Strengths And Weaknesses:**

Strengths:
- Originality: The investigation into GNN's internal layer structure represents a novel contribution. The application of Dirichlet energy to analyze the intrinsic properties of different components within GNN layers is particularly innovative.
- Significance: The identification of opposing smoothing effects between FT and GP operations in GNN layers constitutes an important finding. This insight not only enhances distillation efficiency but also contributes to the broader understanding of GNN architectures, as most GNN models can be analyzed within this framework.
- Clarity: The manuscript is well-written, with the methodology presented in a clear and systematic manner.
- Soundness: The authors provide comprehensive experimental evidence to support their claims.

Weaknesses:

1. The approach requires fixed layer correspondence between teacher and student models. TINED appears unable to generalize to scenarios where teacher and student architectures have significantly different layer counts (e.g., distilling a 4-layer GNN into a 2-layer MLP). An extension of TINED to address this limitation would be valuable.
2. The experimental evaluation focuses exclusively on homogeneous graphs. The performance of TINED on heterophilous graphs remains unexplored.

**Questions For Authors:**

The following questions merit consideration:

1. (Teacher Injection Mechanism): The direct injection of FT parameters from GNNs into MLP layers raises concerns about potential overfitting of the student MLP to the teacher's initialization, particularly if the teacher model is suboptimal. How does the proposed approach address this risk?
2. (Distillation of parameterized GP operations): The handling of parameterized GP operations, such as GAT's attention module, warrants further explanation. Can this teacher knowledge be effectively transferred to the student MLP? This represents an important consideration, as GNN knowledge resides not only in FT layers but also in the parameters of GP operations.
3. (DED on large graphs): For large-scale graphs such as Products, subgraph sampling (\zeta) is employed to approximate Dirichlet Energy. What is the impact of \zeta on the fidelity of DE ratio calculations?
4. (TINED+): The description of the "with graph structure" version of TINED (TINED+) lacks clarity. Does this implementation simply incorporate NOSMOG techniques on top of the base TINED architecture?

**Relation To Broader Scientific Literature:**

The work is appropriately situated within the existing literature:

1. The GNN2MLP pipeline and evaluation benchmarks build upon established research from previous works [1] and [2].
2. The concept of distilling layer-wise teacher structures has been explored in NLP and CV literature. The authors extend this approach to the GNN2MLP distillation domain.
3. The analysis of smoothing effects using Dirichlet energy was introduced in [3], where it was applied to entire non-decoupled GNN layers. This work makes a novel contribution by separately analyzing the FT and GP operations within GNN layers and leveraging their distinct smoothing properties to enhance the distillation process.

[1] Zhang, S., Liu, Y., Sun, Y., & Shah, N (2022). Graph-less Neural Networks: Teaching Old MLPs New Tricks Via Distillation. In International Conference on Learning Representations.
[2] Tian, Y., Zhang, C., Guo, Z., Zhang, X., & Chawla, N. (2023). Learning mlps on graphs: A unified view of effectiveness, robustness, and efficiency. In International Conference on Learning Representations.
[3] Rusch, T. K., Bronstein, M. M., & Mishra, S. (2023). A survey on oversmoothing in graph neural networks. arXiv preprint arXiv:2303.10993.

**Theoretical Claims:**

The theoretical claim regarding the bound on the approximation error of using MLPs to approximate GP (Theorem 4.1) has been verified and found to be mathematically sound.

---

> ### Author Rebuttal · Authors · 2025-04-01
>
> ### Thank you for recognizing the strengths of our work. Here are our responses to your important comments.
>
> > **C1.** Elaboration on deployment with and without graph dependency (line 185-190)
>
> **Response:** When performing inference on a new node with limited connections to a graph, such as a new user joining a social network or a new product being added to an online shopping platform, the node has restricted access to the existing graph structure. In these cases, the graph structure is unavailable to the node  (without graph dependency), so inference relies solely on node features. Conversely, when conducting inference on nodes that have sufficient connections within the graph structure (with graph dependency), it is possible to leverage both node features and graph features. This allows for a graph-dependent inference approach, utilizing the rich information available from the node's connections within the graph.
>
>
> > **C2.** Generalize to student MLP with various layer counts.
>
> **Response:** Thank you for the insightful comment. As shown in Appendix A.3, our approach for decoupled GNNs, such as APPNP in Eq (17), consolidates all graph propagation (GP) operations into a single FC layer in MLP, which can be easily extended to multiple layers. For traditional GNNs like SAGE, our design focuses on capturing layer-to-layer GNN knowledge within MLPs through the proposed techniques, Teacher Injection and Dirichlet Energy Distillation. Consequently, the number of layers in our method is related to the teacher model. One potential way for generalization is to explore layer fusion techniques, where multiple teacher layers are compressed into fewer student layers while maintaining their overall behavior. We will include this discussion in the revised paper.
>
>
> > **C3.** The performance of TINED on heterophilous graphs remains unexplored.
>
> **Response:** Please find the experiments on heterophilous graphs (Squirrel and Amazon-ratings) in the **response to comment C3 of Reviewer qs85**. The results show that TINED and TINED+ outperform existing approaches. As future work, we plan to develop dedicated techniques to further improve the effectiveness of heterophilic GNN distillation.
>
>
>
> > **C4.**  (Teacher Injection Mechanism) How to handle when teacher is suboptimal.
>
> **Response:** We clarify that the goal of knowledge distillation is to create a student model that closely mimics the teacher model, while whether the teacher model is suboptimal or not is not the focus. After injecting teacher parameters on FT operations into the student, the MLP layers in the student are further fine-tuned in Eq. (6) and trained with Dirichlet Energy Distillation and the overall loss in Eq. (8,9). These distillation training steps on the student MLP also mitigates the mentioned risk, and improves student performance.
>
> > **C5.** (Distillation of parameterized GP operations) Explanation on the handling of the parameterized GP operations.
>
> **Response:** As shown in Eq. (4) in Section 4.1 and Eqs. (15, 16, 17) in Appendix A.3, different GNNs have distinct GP operations, such as GAT with attention. Therefore, for these various GP operations, we choose to distill them into fully-connected (FC) layers in MLP, without directly using these GP parameters, but employing carefully designed techniques including layer-wise Dirichlet Energy distillation. On the other hand, the FT operations in different GNNs share a similar formulation that is compatible with FC layers in MLPs, as shown in Eq. (4) and Appendix A.3. Thus, we can directly inject the parameters of FT operations into the student model.
>
>
> > **C6.** The impact of $\zeta$ on large datasets to approximate DE ratio
>
> **Response:**  The parameter $\zeta$ controls the subgraph sampled to estimate the DE ratio, helping to avoid running out of memory on large datasets. We vary $\zeta$ and report the results on the large ogbn-arxiv dataset in **Table A** below. Observe that our TINED maintains stable performance as $\zeta$ changes, and we do not need to consider $\zeta$ larger than 0.8, to reduce computational overhead, while maintaining distillation quality.
>
> **Table A: Vary $\zeta$ on large graph ogbn-arxiv**
> |$\zeta$|0.1|0.3|0.5|0.7|>0.8|
> |-|-|-|-|-|-|
> |ogbn-arxiv|64.31±0.71|64.32±0.91|64.38±0.75|*64.44±0.80*|OOM|
>
>
> > **C7.** The implementation of the "with graph structure" version and NOSMOG.
>
> **Response:** The phrase "with graph structure" refers to conducting inference on nodes that have sufficient connections within the graph, allowing the use of both node features and graph features. In this setting, we use DeepWalk node embeddings for observed nodes, and for unobserved nodes, we use the average positional encodings of their observed neighbors as their encoding, consistent with the ways in NOSMOG. We will include this clarification in the paper.

---

> > ### Comment · Reviewer_bbbm · 2025-04-08
> >
> > I would like to thank the authors for their rebuttal. Since my concerns have been solved, I would like to increase my score to 4.

---

### Official Review · Reviewer_qs85 · 2025-03-13

**Overall Recommendation:** 3

**Summary:**

This work proposes TINED, a method to distill knowledge from teacher GNNs into student MLPs. Extensive experiments show its good performance.

**Claims And Evidence:**

No

**Essential References Not Discussed:**

I am not familiar with this field.

**Experimental Designs Or Analyses:**

No, but the author seems to provide the code. I do not check that.

**Methods And Evaluation Criteria:**

Yes

**Other Comments Or Suggestions:**

na

**Other Strengths And Weaknesses:**

pros:
1. well-written and easy to follow (figures are clear)
2. The topic (GNN distillation) is promising from a practical perspective.
3. The idea is easy and interesting,

cons:
1. I doubt that the method relies highly on hyperparameters. Could the author provide the sensitivity analyses on the datasets mentioned in this paper?
2. I am not very familiar with this field but I remember there are many works focusing on distill GNN into MLP. I hope other reviewers check the novelty of this work.
3. The results seem to rely on homophilic datasets in this paper. Could the author add experiments on heterophilic dataset( eg. Amazon-Ratings, squirrel, pokec, etc).

**Questions For Authors:**

See cons.

**Relation To Broader Scientific Literature:**

na

**Theoretical Claims:**

No

---

> ### Author Rebuttal · Authors · 2025-04-01
>
> ### We appreciate your effort in reviewing our paper. We have conducted extensive experiments to address your key comments. Thank you for your consideration.
>
> > **C1.** Could the author provide the sensitivity analyses on the datasets mentioned in this paper?
>
> **Response:** In our method, the parameter $\eta$ controls the degree of fine-tuning in Eq. (6), while $\beta$ controls the importance of Dirichlet Energy distillation in Eq. (9). We have varied them with  results on  A-computer and Cora datasets in **Tables 4** and **5** of the paper. Below, **Tables A** and **B** present the accuracy results of varying $\eta$ and $\beta$ across more datasets. In **Table A**, as $\eta$ increases across all datasets, TINED exhibits a distinct pattern where performance first improves and then declines, with the best results in italics. This trend underscores the trade off controlled by $\eta$. A similar pattern is observed for $\beta$ in Table B. Note that hyperparameter tuning is essential in machine learning research. Appendix A.7 of the paper details the search space for our parameters.
>
>
> **Table A: Vary $\eta$. Results are averaged over 10 runs with standard deviation**
> |  $\eta$| cora | citeseer | pubmed | a-computer | a-photo |
> |-------------|------|----------|--------|------------|---------|
> | 1e-09 | 73.80±2.56 | 67.20±1.86 | 73.39±2.10 | 72.53±2.17 | 86.06±2.38 |
> | 1e-06 | 73.50±2.48 | 67.19±1.87 | 73.32±1.88 | 72.73±1.99 | 86.35±1.79 |
> | 0.001 | 77.12±0.91 | 67.24±1.83 | 73.69±1.83 | 84.13±1.23 | 87.30±2.08 |
> | 0.01 | 79.26±1.30 | 69.38±1.70 | 75.12±2.32 | *85.17±1.21* | 90.48±1.38 |
> | 0.1 | 81.40±1.69 | 71.52±1.58 | 76.55±2.81 | 83.61±1.67 |*93.97±0.54* |
> | 0.5 | 82.01±1.64 | *74.57±1.42* | *77.10±2.15* | 83.23±1.29 | 93.51±0.60 |
> | 1.0 | *82.61±1.58* | 73.57±1.39 | 76.65±2.77 | 82.85±0.91 | 93.37±0.51 |
> | 10.0 | 78.85±1.63 | 73.56±1.48 | 75.61±2.73 | 70.63±4.87 | 89.53±0.78 |
>
>
>
> **Table B: Vary $\beta$**
> | $\beta$ | cora | citeseer | pubmed | a-computer | a-photo |
> |-------------|------|----------|--------|------------|---------|
> | 1e-09 | 81.33±1.49 | 73.39±1.28 | 76.63±2.42 | 84.70±1.15 | 93.41±0.61 |
> | 1e-06 | 81.64±1.71 | 73.39±1.30 | 76.28±2.76 | *85.17±1.21* | 93.48±0.65 |
> | 0.001 | 81.64±1.57 | 73.39±1.31 | 76.51±2.64 | 84.80±1.21 | 93.65±0.68 |
> | 0.1 | 81.71±1.59 | 73.65±1.38 | *77.10±2.15* | 84.70±0.99 | *93.97±0.58* |
> | 1.0 | *82.61±1.58* | 73.81±1.29 | 75.84±2.66 | 84.29±1.08 | 86.59±5.66 |
> | 10.0 | 80.92±2.15 | *74.57±1.42* | 71.57±2.76 | 71.35±6.30 | 78.00±3.19 |
>
>
> > **C2.** I am not very familiar with this field but I remember there are many works focusing on distill GNN into MLP. I hope other reviewers check the novelty of this work.
>
> **Response:** In other reviews, the reviewers acknowledged the comprehensive experimental evaluations compared to existing methods, as well as the originality and significance of our methods. Thank you.
>
> > **C3.** Could the author add experiments on heterophilic dataset.
>
> **Response:** As suggested, we have conducted new experiments on representative heterophilic datasets, Squirrel and Amazon-ratings under the production setting. In **Tables C(1)** and **C(2)** with and without graph dependency, our  TINED and TINED+ outperform existing approaches under all settings, often with a significant margin.
> Note that our work, as well as existing methods, does not explicitly focus on graphs with heterophily. As future work, we plan to conduct an in-depth investigation and develop dedicated techniques to further improve the effectiveness of heterophilic GNN distillation.
>
>
> **Table C(1): *prod* (*ind* & *tran*) setting on heterophilic dataset (without graph dependency)**
>  | Data | Eval | SAGE | KRD |FFG2M|GLNN | GLNN* | TINED |
> |-------|-----|------|- |-|------|-------|--------|
> | Squirrel | *prod* | 35.47 |38.34 |37.16| 39.90 | 39.70 | **41.95** |
> | | *ind* | 41.44±4.66 |42.00±4.78 |42.11±4.77| 44.89±5.67 | 45.00±4.56 | **46.89±5.23** |
> | | *tran* | 33.98±1.66 |37.43±3.34 |35.93±3.36| 38.65±1.09 | 38.37±1.02 | **40.72±1.36** |
> | Amazon-ratings | *prod* | 47.55 | 50.33 |49.56| 49.87 | 49.41 | **50.70** |
> | | *ind* | 47.45±1.48 |47.55±0.97 |47.68±1.00| 47.72±1.00 | 47.71±1.14 | **49.02±1.02** |
> | | *tran* | 47.58±0.48  | 51.03±1.73   |50.03±1.50| 50.41±0.45 | 49.84±0.39 |**51.12±0.44** |
>
> **Table C(2): *prod* (*ind* & *tran*) setting on heterophilic dataset (with graph dependency)**
>  | Data | Eval | NOSMOG |NOSMOG* | TINED+ |
> |-------|-----|---------|--------|-------|
> | Squirrel | *prod* | 38.17 | 39.43 | **40.89** |
> | | *ind* | 45.44±4.75 | 44.33±3.24 | **46.78±3.80** |
> | | *tran* |  36.35±1.69 |38.20±1.59 | **39.42±1.05** |
> | Amazon-ratings | *prod* | 47.86 | 48.80 | **50.42** |
> | | *ind* |47.47±1.45 | 48.46±1.37 | **49.51±1.54** |
> | | *tran* | 47.96±0.31 | 48.88±0.56 | **50.65±0.56** |

---

### Decision · Program_Chairs · 2025-05-01

**Decision:**

Accept (poster)

**Comment:**

This paper considers the problem of distilling a trained GNN into a cheaper (and often even node-wise) MLP, in order to accelerate downstream inference. In contrast to some prior approaches, the authors perform this layer-by-layer; for every GNN layer, graph propagation is replaced by an MLP regularized with Dirichlet Energy Distillation, and the fully-connected feature transformation is copied over verbatim and then fine-tuned end-to-end to match soft labels from the GNN teacher. This framework is then benchmarked across various datasets, showing excellent results, often outperforming the original GNN.

After the rebuttal period, all four reviewers voted to accept this work, two of them even making a strong recommendation to do so. Reviewers generally praised a well-motivated design, deep analysis, and above all exceptionally good empirical performance. While TINED-produced models are not faster than those obtained from other distillation methods, this is expected as all of those produce very simple models with already rather minimal inference time requirements. Finally, reviewers were also generally happy about the breadth of experiments and reasonable choice of up-to-date baselines.

Given that all four reviewers voted to accept this work, and some championed it, I am inclined to side with the unanimous reviewer decision, and also recommend acceptance.